



# The contrasting response of outlet glaciers to interior and ocean forcing

John Erich Christian[1], Alexander Robel[2], Cristian Proistosescu[3,4,5], Gerard Roe[1], Michelle Koutnik[1], and Knut Christianson[1]

[1]Department of Earth and Space Sciences, University of Washington, Seattle, Washington, USA
[2]School of Earth and Atmospheric Sciences, Georgia Institute of Technology, Atlanta, Georgia, USA
[3]Joint Institute for the Study of the Atmosphere and the Ocean, University of Washington, Seattle, Washington, USA
[4]Department of Atmospheric Sciences, University of Illinois, Urbana-Champaign, Illinois, USA
[5]Department of Geology, University of Illinois, Urbana-Champaign, Illinois, USA

**Correspondence:** John Erich Christian (jemc2@uw.edu)

**Abstract.** The dynamics of marine-terminating outlet glaciers are of fundamental interest in glaciology, and affect mass loss from ice sheets in a warming climate. In this study, we analyze the response of outlet glaciers to different sources of climate forcing. We find that outlet glaciers have a characteristically different transient response to surface-mass-balance forcing applied over the interior than to oceanic forcing applied at the grounding line. A recently developed reduced model represents

outlet glacier dynamics via two widely-separated response timescales: a fast response associated with grounding-zone dynamics, and a slow response of interior ice. The reduced model is shown to emulate the behavior of a more complex numerical model of ice flow. Together, these models demonstrate that ocean forcing first engages the fast, local response, and then the slow adjustment of interior ice, whereas surface-mass-balance forcing is dominated by the slow interior adjustment. We also demonstrate the importance of the timescales of stochastic forcing for assessing the natural variability of outlet glaciers, highlighting

that decadal persistence in ocean variability can affect the behavior of outlet glaciers on centennial and longer timescales. Finally, we show that these transient responses have important implications for: attributing observed glacier changes to natural or anthropogenic influences; the future change already committed by past forcing; and the impact of past climate changes on the preindustrial glacier state, against which current and future anthropogenic influences are assessed.

## 1 Introduction

Marine-terminating outlet glaciers drain large portions of the Greenland and Antarctic ice sheets, conveying ice from interior catchments to the ocean. Their dynamic response to a changing climate is a critical component of projections of sea-level rise (e.g., Aschwanden et al., 2019), and dramatic increases in discharge over recent decades have been observed in Greenland (e.g., Howat et al., 2008; Moon and Joughin, 2008; Moon et al., 2012) and Antarctica (e.g., Pritchard et al., 2009; Miles et al., 2013; Cook et al., 2016). Ocean forcing is thought to play a major role in observed change (e.g., Nick et al., 2009; Joughin

et al., 2012a; Straneo and Heimbach, 2013; Jenkins et al., 2016). Increased melt and runoff driven by atmospheric warming is also a major component of mass loss in Greenland (e.g., van den Broeke et al., 2009).



Yet despite the clear signals of mass loss from the Greenland and Antarctic ice sheets (Shepherd et al., 2018), observed changes in terminus positions, velocities, and ice thickness vary widely among marine-terminating outlet glaciers, especially in Greenland (e.g., Moon et al., 2014; Csatho et al., 2014). This makes it difficult to establish consistent links between forcing

and response, which are critical for projections of future glacier and ice sheet change. Some reasons for the heterogeneity in glacier change have become clearer. For example, localized features in bedrock and fjord geometry can control differences in terminus retreat (Catania et al., 2018) and inland dynamic thinning (Felikson et al., 2017) for individual glaciers. Additionally, regional variations may be associated with differences in surface melt and subglacial hydrology (Moon et al., 2014), or the ocean waters with which the glaciers are in contact (Straneo et al., 2012). However, catchment-specific factors continue to pose

a challenge where observations are limited, as well as for ice-sheet-wide simulations. Further observations will continue to be critical, but this heterogeneity between outlet glaciers is also a strong motivation to investigate general principles at the same time, which can be expected to apply to a wide range of settings.

In this study, we focus on a principle common to all outlet glaciers: climate forcing predominantly comes either from the atmosphere, via changes to the surface mass balance in the interior catchment, or from the ocean via changes to ice

discharge at the marine margin. Our approach is to conduct idealized model experiments that isolate key physical principles of transient glacier responses to climate. We use a recently-developed reduced model (Robel et al., 2018) and an established numerical ice-flow model (Pollard and DeConto, 2012) as complementary tools. The reduced model yields simple physical and mathematical interpretations of the numerical model's response to forcing. Additionally, the reduced model can efficiently generate ensembles of responses for statistical analyses.

We will focus exclusively on stable geometries. The marine-ice-sheet instability (e.g., Weertman, 1974; Schoof, 2007) is of course a critical consideration for some catchments, especially in West Antarctica. However, understanding how a physical system approaches, or fluctuates around, a long-term equilibrium is a core analytical approach. This requires that we start with stable systems, but the resulting insights can often be applied to unstable regimes (e.g., Robel et al., 2019). The following section describes experiments that illustrate the fundamental system dynamics, and establish the key differences between ocean

and interior forcing. We then present three cases that illustrate the implications of these principles for interpreting observations and predicting the future behavior of outlet glaciers.

## 2    Part 1: Dynamical responses to ocean and interior forcing

### 2.1    A simple outlet glacier system

Before describing the dynamic models, it is useful to begin with the geometry and basic flux-balance arguments of the system

we are investigating. We consider an idealized stable outlet glacier, a schematic of which is presented in Fig. 1a (see also Schoof, 2007; Robel et al., 2018). Ice enters as snow accumulation over the interior surface, flows from the interior towards the ocean on a forward-sloping (prograde) bed, and exits the system where it reaches flotation at the grounding line. Beyond this point, floating ice is assumed to calve into icebergs or melt due to contact with the ocean.





Ice flux across the grounding line ($Q_g$) has long been known to be a sensitive function of local ice thickness (e.g., Weertman,
1974; Thomas, 1979; Lingle, 1984; Schoof, 2007). This function can be represented in general form as

$$Q_g = \Omega h_g^{\beta}, \tag{1}$$

where $h_g$ is ice thickness at the grounding line, and $\Omega$ and $\beta$ depend on ice dynamics near the grounding line. A floating ice
shelf, tongue, or friction from valley sidewalls typically provides buttressing. This can be modeled explicitly, or analytically
represented in $\Omega$ and $\beta$. Typically, $\beta \geq 1$, and this nonlinearity has been shown to depend on assumptions about the basal
rheology near the grounding line (Schoof, 2007; Tsai et al., 2015) and the characteristics of a buttressing ice shelf (Haseloff
and Sergienko, 2018). $h_g$ is simply determined by a flotation criterion: let $b(x)$ be the bed elevation relative to sea level (i.e.,
negative at the grounding line). Then, for a glacier of length $L$, $h_g$ is:

$$h_g(L) = -\frac{\rho_w}{\rho_i} b(L), \tag{2}$$

where $\rho_w$ and $\rho_i$ are the densities of seawater and ice, respectively.
In this study, we consider the grounding line the boundary of the outlet glacier system. That is, we consider floating ice a part
of the boundary condition for the system's output flux ($Q_g$). In equilibrium, $Q_g$ equals the surface mass balance integrated over
the entire catchment (neglecting basal and englacial melt). Climate changes can perturb this flux balance in two distinct ways:
the atmosphere can affect mass balance over the interior surface, or the ocean can modulate ice discharge at the grounding
line (e.g., via a change in buttressing). In either case, Eqs. (1) and (2) reveal the basic system response to an imbalance: on a
prograde bed, the terminus retreats into shallower water to decrease ice flux out of the system, or advances into deeper water
to increase flux out of the system. The grounding-line migration needed to restore equilibrium depends strongly on bed slope,
and the degree of nonlinearity in $Q_g$ (i.e., via $\beta$).

## 2.2 Flowline model

To simulate the dynamics of this outlet-glacier system, we begin with a 1-D (flowline) version of the ice-sheet model developed
by Pollard and Deconto (2012; hereafter PD12).

The evolution of local ice thickness ($h$) at a grid point reflects the balance of mass exchange at the surface and horizontal
ice-flux divergence:

$$\frac{\partial h}{\partial t} = S - \frac{\partial(\bar{u}h)}{\partial x}, \tag{3}$$

where $S$ is the local surface mass balance and $\bar{u}$ is depth-averaged horizontal ice velocity. The velocity profile has contributions
from stretching, shear, and basal sliding, which are summarized as follows. In general, a sloping ice surface ($\partial s/\partial x$) creates a
driving stress,

$$\tau_d = \rho_i gh(\partial s/\partial x) \tag{4}$$

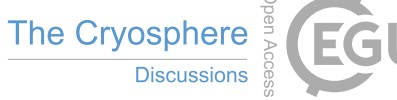

where $\rho_i$ is ice density and $g$ is acceleration due to gravity. Longitudinal stretching ($\partial u/\partial x$) is represented by the shallow-shelf approximation, where stretching and basal drag ($\tau_b$) balance driving stress:

$$\frac{\partial}{\partial x}\left(2hA^{(-1/n)}\left|\frac{\partial u}{\partial x}\right|^{(1/n-1)}\frac{\partial u}{\partial x}\right) - \tau_b = \tau_d. \tag{5}$$

Here, $A$ and $n$ are the Glen-type coefficient and exponent (e.g., Glen, 1955). Internal shear deformation ($\partial u/\partial z$) is represented by the shallow-ice approximation. At a depth $h - z$, this is:

$$\frac{\partial u}{\partial z} = 2A\tau_e^{n-1}\rho_i g(h-z)\frac{\partial s}{\partial x} \tag{6}$$

where $\tau_e^2 = (1/2)\tau_{ij}\tau_{ij}$ is a scalar effective stress and $\tau_{ij}$ is the deviatoric stress tensor. Finally, sliding velocity is given by a Weertman-type power-law relationship:

$$u_b = (\tau_b/C)^{1/m} \tag{7}$$

where $C$ is a friction coefficient and $m$ is the sliding exponent. Here, $m$ is the inverse of the flow exponent $n$, but note that this convention is flipped in some texts. The PD12 model model uses a combination of the shallow-ice and shallow-shelf approximations to solve for $\partial u/\partial z$ and $\partial u/\partial x$, respectively. In this study, however, we use parameters that yield flow dominated by longitudinal stretching. We refer the reader to PD12 for further description of this hybridization.

A crucial simplification in the PD12 model is that flux across the grounding line is parameterized in the form of Eq. (1). In PD12 and this study, $\Omega$ and $\beta$ are taken from the analytical solution of Schoof (2007):

$$\Omega = \left[A(\rho_i g)^{n+1}(\Theta(1-\rho_w/\rho_i))^n(4^nC)^{-1}\right]^{\frac{1}{m+1}} \tag{8}$$

$$\beta = \frac{m+n+3}{m+1}. \tag{9}$$

$\Theta$ is a buttressing factor between 0 and 1, and all other parameters are defined as above. The PD12 model calculates grounding line flux based on a thickness ($h_g$) that is linearly interpolated from the height above flotation of the last-grounded and first-floating grid cells, to the point where flotation is reached. Although the PD12 model is typically run on coarse grids ($\mathcal{O} \sim 1$–10 km) for continent-scale simulations over many millennia, we use a grid of 100 m to better resolve the details of grounding line variations. We use the same interpolation scheme to estimate the sub-grid grounding line position. Finally, while the PD12 model does simulate a floating ice shelf, its dynamics are not important to our analyses as its buttressing effect is parameterized via Eq. (8).

Figure 1b shows the steady-state profile for an idealized outlet glacier simulated with the flowline model. The domain begins at an ice divide and thus has a zero-flux lateral boundary condition. The bed has an elevation of $-100$ m at the divide, and a constant prograde slope ($b_x$) of $-2 \times 10^{-3}$. Surface mass balance ($S$) is 0.5 m yr$^{-1}$ ice equivalent, assumed to be spatially uniform. Additional parameters are given in Table 1. The glacier has an equilibrium length (ice divide to grounding line) of





$\sim$ 185 km, a maximum thickness of $\sim$ 1580 m at the divide, and a thickness of $\sim$ 526 m at the grounding line. These scales are comparable to many outlet glacier catchments in Greenland, though direct comparisons are limited as the 1-D flowline cannot capture the flow convergence of many catchment geometries, and we emphasize that we are not simulating a particular outlet

glacier. We discuss two additional geometries for comparison in Sect. 3.1.

## 2.3 Response to forcing

We begin by comparing the flowline model's response to forcing either from surface-mass-balance changes in the interior or ocean forcing at the terminus. Surface-mass-balance anomalies are assumed to be spatially uniform. We represent ocean forcing very simply by perturbing the grounding-line-flux coefficient, $\Omega$. This broadly represents changes in the buttressing provided

by an ice shelf or fjord walls, driven by anomalous melting or calving. In principle this could be targeted via $\Theta$ in Eq. (8), and other analytical formulations for $\Omega$ explicitly represent a buttressing ice shelf (Haseloff and Sergienko, 2018). Perturbing these parameters might be more realistic, but we focus on $\Omega$ so that we can very generally represent flux perturbations, which may in reality result from a host of ice-ocean interactions. Our primary interest is in how glacier dynamics respond to each forcing type, and representing ocean forcing in this way makes comparison straightforward. For example, a fractional change

in $S$ (surface mass balance) constitutes a flux anomaly with the same initial magnitude as the same fractional change in $\Omega$.

Figure 1c shows the length response to step forcings at time $t = 0$. The step is a 20% decrease in $S$ or a 20% increase in $\Omega$ (i.e., more discharge for a given $h_g$), and in both cases, forces the terminus to retreat into shallower water to reduce the output flux. Note that the responses to changes in $S$ and $\Omega$ do not converge to the same final length. The equilibrium sensitivities differ because of different nonlinearities in the input and output fluxes ($S \times L$ and $Q_g$) as the system state evolves. However,

our main focus is the marked difference between the initial responses to the step change, highlighted in the inset panel of Fig. 1c. Perturbing $\Omega$ results in a much faster initial retreat (blue curve) compared to perturbing $S$ (orange curve), despite the larger equilibrium sensitivity to $S$. However, this faster retreat rate lasts only the first century or so, and accommodates only $\sim 25\%$ of the total equilibrium response. The remaining retreat occurs at a rate similar to that driven by a change in $S$, which is an asymptotic approach to equilibrium over several millennia. Thus, the length response to $\Omega$ forcing has both a "fast" and "slow"

component, whereas $S$ forcing primarily drives a slow response.

As an alternative to an impulse forcing, we also consider the glacier's response to stochastic variability in either $\Omega$ or $S$. We apply this variability as interannual white noise, which by definition has equal power at all frequencies and no interannual persistence. We apply the exact same white-noise timeseries as either $\Omega$ or $S$ anomalies, but with opposite sign so that the corresponding ice-volume anomalies match. The anomaly timeseries (hereafter $\Omega'$ and $S'$) are scaled to have a standard devi-

ation equal to 20% of the mean values ($\bar{\Omega}$ and $\bar{S}$). Figure 1d shows the resulting length responses. For both types of forcing, the glacier acts as a low-pass filter on the imposed climate anomalies, producing kilometer-scale fluctuations with clear persistence. However, the length anomalies driven by $\Omega'$ have much greater high-frequency content, and greater overall variance, than those driven by $S'$. The high-frequency response is perhaps intuitive, in that forcing applied at the grounding line has an immediate effect on grounding-line position. However, the response to $\Omega'$ also contains millennial-scale fluctuations onto





**Table 1.** Parameters used for flowline and two-stage models.

| Parameter | Value |
|---|---|
| Sliding exponent, $m$ | 1/3 |
| Sliding coefficient, $C$ (Pa m$^{-1/3}$ s$^{1/3}$) | $7.624 \times 10^{6}$ |
| Deformation exponent, $n$ | 3 |
| Deformation coefficient, $A$ (Pa$^{-3}$ s$^{-1}$) | $4.22 \times 10^{-25}$ |
| Buttressing factor, $\Theta$ | 0.7 |
| Seawater density, $\rho_w$ (kg m$^{-3}$) | 1028 |
| Ice density, $\rho_i$ (kg m$^{-3}$) | 917 |
| Surface mass balance, $\bar{S}$ (m yr$^{-1}$) | 0.5 |
| Bed elev. at divide, $b_0$ (m) | $-100$ |
| Bed slope, $b_x$ | $-2 \times 10^{-3}$ |

which the faster variations are superimposed. These slow, wandering excursions are comparable to those driven by $S'$, and, like the multi-millennial adjustment to step changes (Fig. 1c), suggest slow dynamics common to both responses.

Variability is intrinsic to climate, arising from the fundamentally chaotic nature of the natural system. The associated glacier fluctuations are a crucial part of characterizing glacier dynamics, and the implications have been extensively explored for mountain glaciers (e.g., Oerlemans, 2001; Roe, 2011; Roe et al., 2017), and more recently for marine-terminating outlet glaciers
(Robel et al., 2018) and ice streams (Mantelli et al., 2016). Figure 1 suggests a new and intriguing complication for outlet glaciers: the timescale and magnitude of glacier fluctuations depend on whether the climate variability comes in the form of surface mass balance or ocean anomalies. Additionally, the response to step changes (Fig. 1c) suggests that the type of forcing is also relevant for the response to non-stationary climate changes. In the next section, we investigate these contrasting responses using a recently developed reduced model.

**2.4 The two-stage model**

Robel, Roe, and Haseloff (2018; hereafter RRH) developed a simple model for marine-terminating outlet glacier dynamics, derived from ice-flux conservation and constrained by large-scale glacier geometry. RRH showed that this model could accurately emulate a more complex numerical model on the multidecadal and longer timescales on which most glacier variance occurs. A full description and derivation can be found in RRH, but a summary of the model follows.
The RRH model reduces the outlet glacier to a system with two degrees of freedom: $L$, the length from ice divide to grounding line; and $H$, the spatially averaged interior ice thickness (Fig. 2a). The glacier can be conceptualized as a small grounding-zone reservoir with a length $l_{gz} \ll L$ and thickness $h_g$, coupled to a large interior reservoir with thickness $H$ and length $L - l_{gz}$. As such, we will refer to this as the "two-stage model". The geometry is further described by a static bed



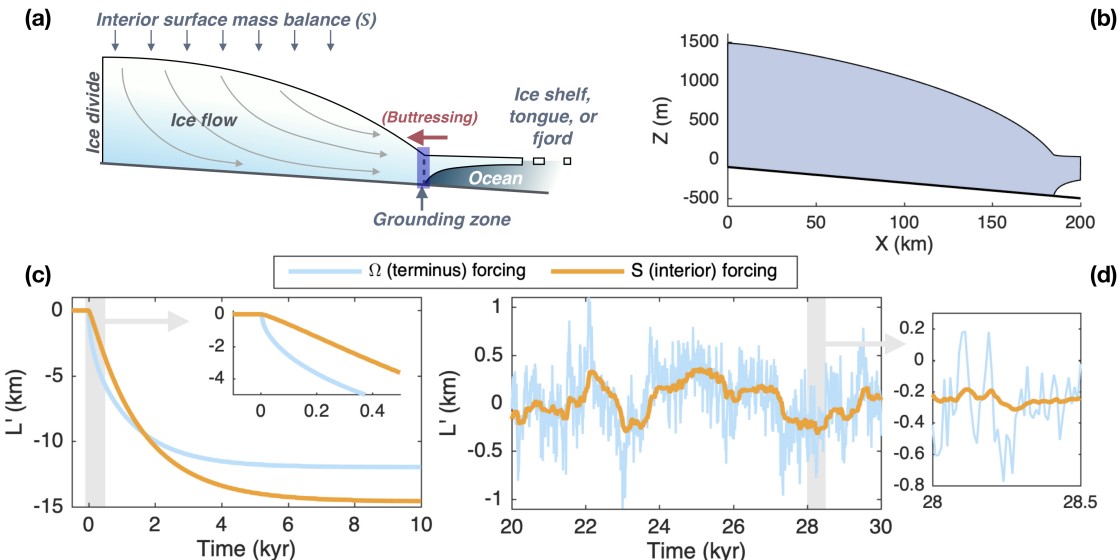

**Figure 1.** Model schematic and response to forcing. **(a)** An idealized marine-terminating glacier system. **(b)** Intial steady-state profile as simulated by the flowline model. **(c)** Length response to 20% step increase in $\Omega$ (blue) and 20% step decrease in $S$ (orange) at $t = 0$. Inset panel zooms into the first 0.5 kyr of response. Note the difference in initial retreat rates depending on the type of forcing. **(d)** Length response to random, normally-distributed interannual variability in $\Omega$ and $S$. Anomalies have a standard deviation of 20% of the mean value, and have opposite signs for $\Omega$ and $S$ so that length anomalies will be correlated for comparison. The response to forcing at the grounding line ($\Omega$) contains much more high-frequency content.

topography with average slope $b_x$. Recall that the ice thickness at the grounding line, $h_g$, is a state-dependent quantity, set entirely by $L$ and $b(x)$ (see Eq. 2).

The two-stage model dynamics are derived by balancing ice fluxes through these linked reservoirs (Fig. 2a). Ice thus enters the system via an accumulation flux ($S \times L$), flows from the interior to the grounding zone ($Q$), and leaves the system as a flux across the grounding line ($Q_g$). A steady state of $H$ and $L$ is one which balances these three fluxes. $L$ ultimately controls the system's input and output fluxes by setting the total catchment area for accumulation, and by controlling $Q_g$ via $h_g$ (Eqs. 1 and 2). Interior dynamic ice flux has the form

$$Q = \left(\frac{\rho_i g}{C}\right)^n \frac{H^\alpha}{L^\gamma}, \tag{10}$$

where $\alpha$ and $\gamma$ can vary depending on the particular processes controlling interior flow. We use $\alpha = 2n+1 = 7$ and $\gamma = n = 3$, consistent with deformation dominated by longitudinal stretching (RRH).

Two coupled equations capture the transient adjustment of the two degrees of freedom, $H$ and $L$, as they relax towards a steady state that balances all three fluxes:

$$\frac{dH}{dt} = S - \frac{Q}{L} - \frac{H}{h_g L}(Q - Q_g) \tag{11}$$





$$\frac{dL}{dt} = \frac{1}{h_g}(Q - Q_g). \tag{12}$$

Note that Eq. (10) captures the nonlinear dependence of ice flux on driving stress (e.g., Eqs. 4–6) and ice thickness, and that
the first two terms of Eq. (11) resemble the continuity equation that governs ice thickness in the flowline model (Eq. 3). The
two-stage model makes significant simplifications, but stems from the same physical principles as the flowline model, and most
other contemporary ice-sheet models (see RRH for further discussion).

RRH also linearized these equations for fluctuations $H'$ and $L'$ about a steady state, $\bar{H}$ and $\bar{L}$. The linear model yields
analytical expressions for a number of important quantities, including two widely separated characteristic response timescales.
With some simplifications detailed in RRH, the fast and slow response timescales ($\tau_F$ and $\tau_S$, respectively) are

$$\tau_F = \frac{\bar{h_g}}{\bar{S}}(\alpha + \gamma + 1 - s_T)^{-1} \tag{13}$$

$$\tau_S = \frac{\bar{H}}{\bar{S}s_T\alpha}(\alpha + \gamma + 1 - s_T), \tag{14}$$

where $s_T = 1 + \frac{\rho_w}{\rho_i}\frac{\beta\bar{b_x}\bar{L}}{h_g}$ is a parameter describing the stability of glacier response based on its geometry and grounding-
line dynamics (via $\beta$). Both response times contain a characteristic thickness divided by mass balance rate, reminiscent of
the canonical mountain glacier response time (Jóhannesson et al., 1989). $\tau_F$ is controlled by $h_g$, and thus the volume of the
system's small reservoir, whereas $\tau_S$ relates to the volume of the interior reservoir via $\bar{H}$. For scales typical of outlet glaciers,
$\tau_F$ is on the order of decades to centuries, whereas $\tau_S$ is on the order of millennia. Thus, the two-stage model approximates
the outlet-glacier system as linked interior- and grounding-zone reservoirs with distinct timescales.

Here we generalize the RRH linearization to simultaneously include perturbations to interior surface mass balance ($S'$) and
grounding-line flux ($Q_g'$; proportional to $\Omega'$). The linearized equations take the form of a two-dimensional dynamical system
with two forcing terms:

$$\frac{\partial}{\partial t}\begin{bmatrix} H' \\ L' \end{bmatrix} = \begin{bmatrix} A_H & A_L \\ B_H & B_L \end{bmatrix}\begin{bmatrix} H' \\ L' \end{bmatrix} + \begin{bmatrix} \bar{L}^{-1}(\bar{H}/\bar{h_g} - 1) \\ \bar{h_g}^{-1} \end{bmatrix}Q_g' + \begin{bmatrix} 1 \\ 0 \end{bmatrix}S'. \tag{15}$$

$A_H$, $A_L$, $B_H$, and $B_L$ are shorthand for the couplings between length, thickness, and flux changes, which are given in the
appendix. This linear system is readily discretized, and can be cast as a two-dimensional autoregressive process (e.g., Box
et al., 2015), which we also present in the appendix.

## 2.5  Model comparison

RRH showed that the two-stage model emulated the response of a different flowline model to interior surface-mass-balance
anomalies. Here, we use the PD12 model and extend the comparison to grounding-line flux perturbations. Although the two-
stage model output includes $H$, we focus our comparison on $L$ anomalies, which are more directly comparable between models.





$H$ is a more heuristic variable in the two-stage model; however, it does govern the evolution of interior fluxes, which we discuss in the next section.

Figure 2 shows output from both models in response to step and stochastic forcings. For clarity, only the linearized two-stage output is shown. The linear and nonlinear responses match almost exactly for the stochastic fluctuations, but as expected,
disagree slightly for step changes; the nonlinear two-stage model is constrained to have the same equilibrium response as the flowline model by Eq. (1). Fig. 2b shows the grounding line retreat following a 20% increase in $\Omega$ (blues) and 20% decrease in $S$ (orange/brown). The two-stage model captures the faster initial response to forcing at the grounding line, with a slightly more pronounced slope break in the first few hundred years.

Figure 2c shows a 5 kyr sample of 100 kyr of length fluctuations driven by white-noise forcing in $S'$ and $\Omega'$. Again, anomalies
have no interannual persistence, are identical except for a sign reversal between $\Omega'$ and $S'$, and have a standard deviation of 20% of $\bar{\Omega}$ or $\bar{S}$. The two-stage and flowline models agree closely for both types of forcing, although the two-stage model slightly underestimates the magnitude of high-frequency fluctuations. Figure 2d shows the autocorrelation function of each length timeseries. This is a measure of the persistence of anomalies, and reveals the memory within a dynamical system. For forcing from $\Omega'$, the autocorrelation drops off rapidly to approximately $0.4$ after a couple of centuries, corresponding to the fast
response time. However, it then requires several millennia to drop below $\sim 0.1$, indicating the persistent influence of the slow timescale. For $S'$ forcing, the autocorrelation is dominated by the slow timescale. The power spectra of length fluctuations are shown in Fig. 2e. Because white-noise forcing has a flat power spectrum, the shapes of the response spectra reveal the glacier's filtering properties. The glacier is a strong low-pass filter for both forcing types, but damps high frequencies even more for $S'$ forcing. The two-stage model underestimates the high-frequency response of the flowline model, but agreement is very good
at the lower frequencies that contain the majority of the variance. Critically, both models capture the clear split between the spectra of $\Omega'$- and $S'$-driven anomalies at frequencies around $10^{-2}$ to $10^{-3}$ $yr^{-1}$, where the models agree quite well.

## 2.6 Interpretations

The two-stage model's agreement with the flowline model suggests that it captures the essential dynamics responsible for the contrasting transient responses to interior and grounding-line flux anomalies. At this point, it is useful to discuss some of the
interpretations enabled by this reduced model.

The linearized two-stage model (Eq. 15) is a dynamical system with two eigenmodes, which, for the stable geometries considered here, are exponentially decaying functions with characteristic times $\tau_F$ and $\tau_S$ (the eigenvalues being $\tau_F^{-1}$ and $\tau_S^{-1}$). The responses of the system's two degrees of freedom ($H'$ and $L'$) are linear combinations of these "fast" and "slow" eigenmodes. However, the two forcing types, $\Omega'$ and $S'$, project onto each mode with different relative weights. Forcing via $S'$
projects almost entirely onto the slow mode. For forcing in $\Omega'$, the fast mode makes a substantial contribution on multi-decadal to centennial timescales. However, the slow mode still dominates the equilibrium response to an impulse, as well as the power spectrum of stochastic fluctuations.

These modes can also be conceptualized as a series of low-pass filters on any forcing time series (e.g., Fig. 2e), and the type of forcing determines the order in which they are applied. Forcing over the interior ($S'$) is first filtered by the slow mode, and



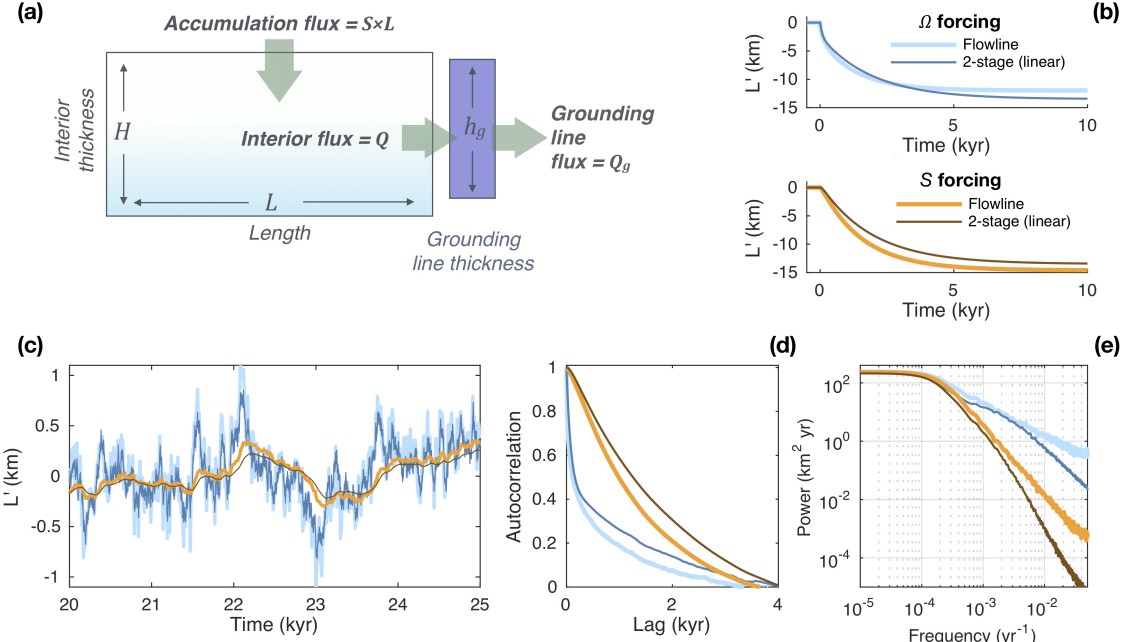

**Figure 2.** Comparing transient behavior of the flowline and two-stage models. **(a)** Schematic of the two-stage model. **(b)** Comparisons of flowline and linearized two-stage model length responses to step changes in $\Omega$ and $S$. The thinner lines show the two-stage output. **(c)** Length anomalies in response to white-noise forcing of both types. As in Fig. 1, anomalies have standard deviation of 20% of the mean value and have opposite signs. 5 kyr of the 100 kyr model runs are shown. **(d)** Autocorrelation function for stochastic length fluctuations. **(e)** Power spectral density for length fluctuations, which is estimated throughout this study via Welch's method with windows of 1/16 the timeseries length, and 50% overlap. Note the clear split between fluctuations driven by $\Omega'$ and $S'$ at millennial and shorter timescales. This split is robust across models. The flowline model has more high-frequency power for both types of forcing, but note that the spectral power is orders-of-magnitude lower at such high frequencies.

the fast mode then operates on interior flux anomalies whose high-frequency content has already been strongly damped; there is little additional filtering to do. In contrast, if forcing is applied at the grounding line ($Q_g'$ via $\Omega'$), the fast mode responds to unfiltered anomalies and the grounding-line position exhibits a fast initial response before engaging the slow mode in the interior. Figures 1 and 2 suggest that the response to $S'$ could be reasonably approximated by the adjustment of a single multi-millennial mode. However, we stress that the response to $\Omega'$ is *not* simply the response of a single multi-decadal mode, because

the grounding zone is coupled to the slower, but ultimately more sensitive interior reservoir.

   While these mathematical interpretations may seem abstract, it is helpful to remember that the two-stage model was derived from mass conservation, and that the linear response times (eigenmodes) reflect the large-scale glacier geometry. Because a glacier is a Stokes (i.e., non-inertial) fluid driven by potential-energy gradients, the large-scale glacier geometry must reflect the aggregate dynamics by which the system seeks flux balance. This relationship between geometry and dynamics is another





way to interpret why $\Omega'$ and $S'$ forcings project differently onto the fast and slow modes: these flux imbalances have different geometries, and thus the geometry and dynamics of the transient responses must also be distinct.

To help illustrate this, Fig. 3 tracks the evolution of fluxes following step changes in $S$ and $\Omega$. Five fluxes are shown for the flowline model: over the interior surface ($S \times L$), across the grounding line ($Q_g$), and at conceptual flux gates located 5, 25, and 50 km up-glacier of the grounding line (Fig. 3a). For the nonlinear two-stage model, $S \times L$, $Q$, and $Q_g$ are shown (Fig.

3b). In all cases, fluxes are normalized by their initial and final equilibria. A change in surface mass balance has an immediate, spatially distributed tendency on interior thickness, which slowly alters driving stresses and thus fluxes throughout the interior. Anomalous ice flux arrives at the grounding zone, driving advance or retreat, which then modifies $Q_g$ according to Eq. (1). Most of the disequilibrium during the transient response is between surface mass balance and interior fluxes, while $Q_g$ can quickly keep pace with flux anomalies from the interior. In the two-stage model, $Q_g$ keeps up on the fast timescale. In contrast,

a perturbation to $Q_g$ (here, via $\Omega$) is highly localized and first creates a large disequilibrium between $Q_g$ and interior flux. The increase in $Q_g$ causes the grounding zone to drain, drawing in ice from the interior, which transfers the disequilibrium from $Q$ and $Q_g$, to $Q$ and $S \times L$. Both models capture this transfer, and the flowline model flux gates show that it gradually propagates up from the grounding zone. The ensuing drawdown of the interior reservoir (the slow mode in the two-stage model) brings interior fluxes back down, and again, $Q_g$ must follow via grounding line retreat. Although the flowline model captures

a more realistic and spatially distributed response, the two-stage model contains enough geometric information to emulate the basic sequence of flux anomalies. Note that more discretized interior reservoirs could be added to the two-stage model, which would eventually approach the form of the flowline model. However, a single reservoir could not capture the essential transient response. The distinct stages of the response to a perturbation in $Q_g$, borne out in both models, show that a two-mode framework is useful for understanding the response to forcing applied at the grounding zone.

## 2.7 Persistence in variability

In the previous sections, we imposed stochastic variability with a flat power spectrum (i.e., white noise) because it allowed us to identify the influence of forcing type and model physics across all relevant frequencies. In reality, surface-mass-balance and ocean variability may exhibit different power spectra. Climate variables associated with the atmosphere (such as surface mass balance) often have little interannual memory (e.g., Medwedeff and Roe, 2017), but the ocean's thermal inertia can

introduce persistence (e.g., Hasselmann, 1976). While the comparisons above show that interior and terminus forcing elicit different glacier responses due to the distinct geometries of the respective flux anomalies, we must also consider the possibility of climatic persistence in order to fully characterize an outlet glacier's response to natural climate variability. Roe and Baker (2016) showed that persistence in surface mass balance variability amplified the length fluctuations of mountain glaciers; we can expect similar principles to be especially relevant to outlet glaciers if they are sensitive to decade-scale ocean variability.

We consider four types of synthetic forcing with different persistence characteristics: (1) white noise; (2) and (3) first-order autoregressive (AR-1) processes with persistence timescales ($\tau_{AR1}$) of 4 and 20 yr, respectively; and (4) power-law persistence. These are all plausible models for the sort of persistence that may affect outlet-glacier forcing. For example, an AR-1 process with a 4 yr memory is characteristic of sea-surface temperature anomalies in the North Atlantic (e.g., Deser et al., 2003),





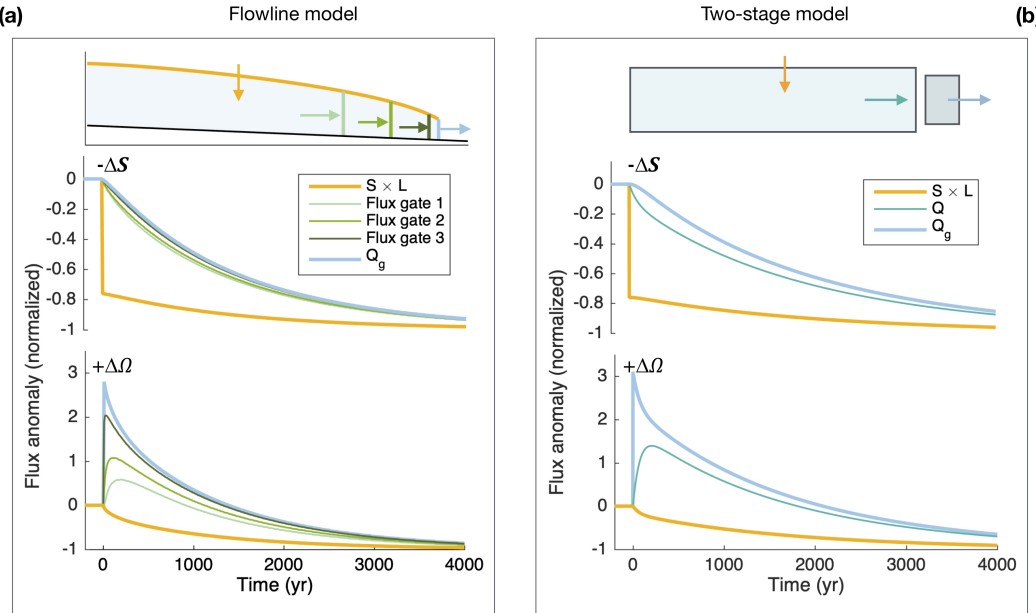

**Figure 3.** Flux changes following interior vs. grounding-zone forcings. (a) Flowline model flux anomalies to perturbations in $S$ and $\Omega$. The schematic (top) shows the flux gates considered. Below, normalized flux responses to the perturbations are shown for each flux gate. (b) as for (a), but for the two-stage model, which has three flux variables. The two-stage model's interior flux variable ($Q$) captures the basic tendencies of interior fluxes in the flowline model. In both cases, the response to a perturbation in $\Omega$ clearly shows the two-timescale nature of outlet glacier dynamics.

while the 20 yr timescale would better correspond to subsurface anomalies (e.g., Gwyther et al., 2018; Jenkins et al., 2018).
Additionally, the continuum of surface-temperature variability from interannual to multimillenial timescales has been described with a power-law spectrum (e.g. Huybers and Curry, 2006).

We generate timeseries following Percival et al. (2001) and Roe and Baker (2016), first defining the desired spectral properties in frequency space, and then applying the same set of random phases to each case so that the resulting anomalies are correlated in the time domain. The power spectrum of a discrete AR-1 process as a function of frequency $f$, is

$$P_r = P_0\big(1 + r^2 - 2r\cos(2\pi\Delta t f)\big)^{-1}, \tag{16}$$

where $P_0$ scales the total variance and $r$ is the auto-correlation at a lag of $\Delta t$ (here, 1 yr). The memory timescale of the process is related to $r$ by $\tau_{AR1} = \Delta t/(1-r)$. Power-law noise has a spectrum defined by

$$P_\nu = P_0\big(f_0/f\big)^\nu, \tag{17}$$

where $f_0$ is the highest sampled frequency. The power increases continually out to low frequencies, where $\nu$ is the slope of
the spectrum in log-log space. We use $\nu = 0.5$, consistent with $\nu \sim 0.5$ to 1 identified by Huybers and Curry (2006) in a large collection of paleoclimate records.





We normalize the forcing timeseries so that they all have the same variance. This ensures that, for a given choice of $S'$ or $\Omega'$ forcing, differences in glacier response are due only to differences in persistence. Figure 4a shows 200 yr samples of the $10^5$ yr synthetic forcing timeseries, and Fig. 4c shows their power spectra. The AR-1 processes (reds) have clear persistence in the time domain. The low-frequency content of power-law noise (blue) is hard to discern at short timescales, but the spectra show that it has similar power to the AR-1 processes at millennial timescales (and, again, identical total variance). Power-law variability is thus difficult to constrain from century-scale instrumental records alone (e.g., Percival et al., 2001). At a certain point, the spectrum of climate variability runs into timescales where paleoclimate reconstructions, rather than synthetic noise or instrumental observations, would be a more relevant choice for understanding natural glacier variability; we will return to this point in section 3.3.

Figure 4b shows $10^3$ yr samples of the resulting length responses for the two-stage model. The effects of persistence are dramatic: for both $S'$ or $\Omega'$ forcing, 4 yr AR-1 persistence more than doubles $\sigma_L$ compared to length fluctuations driven by white forcing, and both 20 yr AR-1 and power-law persistence cause a $\sim$5-to-6-fold increase in $\sigma_L$. This is consistent with the approximate proportionality between $\sigma_L^2$ and $\tau_{AR1}$ predicted by theory (see, e.g., Roe and Baker, 2016; Robel et al., 2019).

The power spectra of the forcings and responses show why persistence has such a strong effect. The response to white forcing reveals the system's sensitivity across all timescales (Fig.4d), because the input has equal power at all frequencies. The glacier response is set by its internal dynamics and the forcing type ($S'$ or $\Omega'$). Any forcing timeseries must be filtered by the same dynamics, and if the forcing has persistence, some of its spectral power is shifted toward lower frequencies where glacier sensitivity is higher. Thus, the shape of the response spectrum (Fig. 4e) reflects both the frequency content of the forcing (Fig. 4c), and the frequency dependence of the glacier dynamics, or, in other words, the spectral shape of the glacier filter (Fig. 4d).

The practical takeaway is that climatic persistence increases the total variance of length fluctuations. When combined with the finding that terminus-flux anomalies excite the fast mode of response more than surface-mass-balance anomalies, the implication is that ocean variability—which tends to have persistence—may drive much larger terminus fluctuations than surface-mass-balance variability.

Figures 1–3 demonstrate that marine-terminating outlet glaciers have different transient responses to interior and terminus forcing, because of how the fast and slow modes respond in each case. In the next sections, we examine the consequences for three key issues: the committed response to forcing, the attribution of an observed change, and a glacier's memory of past climate changes. The relative roles of ocean and surface-mass balance-forcing will, of course, vary widely among individual glaciers. Rather than conducting simulations of specific settings, we will use our simplified model framework to outline the general implications and assess how the combination of fast and slow dynamics applies to each question.

## 3   Part 2: Implications

### 3.1   Committed Change

Any system with a non-zero response time will lag applied forcing. "Committed change" refers to the total response such a system would need to undergo to attain equilibrium with the current level of forcing. In the context of a warming climate,



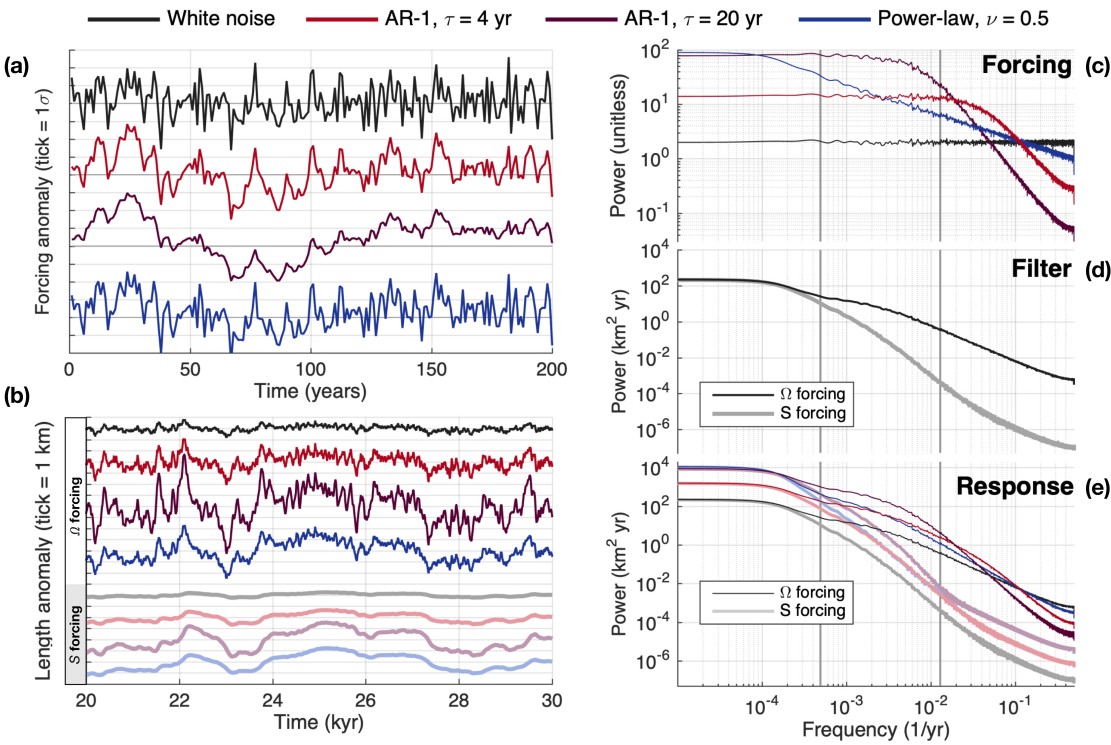

**Figure 4.** The effect of climatic persistence on glacier fluctuations. **(a)** Segments of synthetic anomalies with different levels of persistence: white noise (black); 4 yr AR-1 (red); 20 yr AR-1 (maroon); and power-law ($\nu = 0.5$; blue). Timeseries shown are dimensionless and offset for clarity, with vertical ticks of $1\sigma$. **(b)** The glacier-length variations generated by the two-stage model, when the synthetic anomalies are applied as $\Omega'$ (dark lines) and $S'$ (pastel lines). Timeseries are again offset, with vertical ticks of 1 km. **(c)** Power spectra of the synthetic climate forcings. For reference, vertical lines correspond to $1/\tau_F$ and $1/\tau_S$. **(d)** Spectra of the glacier's response to white noise, which illustrates the glacier's sensitivity as a function of frequency, and thus its temporal filtering properties. This depends on whether forcing is applied at the terminus (black) or interior (gray). **(e)** Spectra of glacier response to each type of synthetic variability, which depend on the spectra of the forcing and glacier's filtering properties. Forcing with persistence has more power at low frequencies where glacier sensitivity is highest, increasing the overall glacier variance.





committed change is an important lower bound on future change that is independent of future emissions scenarios. It has long been recognized that surface temperatures lag $CO_2$ forcing due to the ocean's thermal inertia (e.g., Hansen et al., 1985; Wigley and Schlesinger, 1985). In turn, other aspects of the climate system respond to warming with their own lags, including mountain glaciers (e.g., Christian et al., 2018) and global sea level (e.g., Meehl et al., 2005; Levermann et al., 2013). Industrial-era climate forcing began in the 19th century (e.g., Stocker et al., 2013; Abram et al., 2016), and so the millennial response

times (i.e., Eq. 14) of outlet glaciers imply a severe disequilibrium with current climate and a large committed change, even with no additional forcing.

To illustrate the current committed change of outlet glaciers, we use the two-stage model and follow the framework of Christian et al. (2018), using idealized glacier geometries and forcings. Disequilibrium, and thus committed retreat, is defined in reference to an "instaneous equilibrium" response, which is the state at which the glacier would be in equilibrium with

climate at any given time. This is governed largely by the bed geometry and climate forcing. An idealized geometry (e.g., uniform bed slope and width) makes for a simple equilibrium response, but the physical principle of course also holds for more complex systems. For an idealized outlet glacier, the instantaneous equilibrium is the grounding-line position ($L$) that yields $Q_g = S \times L$.

We consider ramp forcing scenarios where $S$ decreases, or $\Omega$ increases, by 30% from 1880 to 2020 CE. The forcing is fixed

at 2020, and the glacier is allowed to relax towards a new equilibrium. Figure 5 shows the responses of three glaciers with varied response times: (1) $\tau_F, \tau_s \sim (76, 2000)$ yr; (2) $\tau_F, \tau_s \sim (56, 1200)$ yr; and (3) $\tau_F, \tau_s \sim (140, 4600)$ yr. The parameters for each glacier are provided in Table 2. Dashed lines show the instantaneous equilibrium responses, which indicate the total committed response at any given time.

The most basic result is that, in all cases, the transient response is a small fraction ($\sim 1$–$23\%$) of the equilibrium response

as of 2020. The disequilibrium for $S$ forcing is particularly severe, as pointed out previously by RRH. Although changes in surface mass balance are already a major component of overall mass loss in some settings, basic glacier dynamics require that $S$ anomalies also have an impact on the grounding-line position, which is, as yet, essentially unrealized. Perturbations in $\Omega$ elicit a faster response over the industrial era, but Fig. 5a–c shows that both types of forcing have a long-term commitment associated with the slow drawdown of interior ice.

The "slow response" (interior thinning → reduced interior fluxes → grounding-line retreat) comprises the majority of the response to $S$ forcing, but it is important to remember that it is also the second stage of the response to terminus forcing, and it dictates the total committed change. Following recent observed increases in ice discharge, Price et al. (2011) and Goldberg et al. (2015) examined the committed responses of several outlet glaciers in Greenland and Antarctica, respectively. These studies found substantial "dynamic" commitments associated with thinning and elevated fluxes, although they focused on

near-term sea-level rise and did not project past 2100. Our flux-balance perspective is a complementary view of committed change, and shows that dynamic thinning necessarily drives additional retreat on long timescales as a consequence of reduced interior fluxes. The slow equilibration of the interior would thus be a critical part of determining the total commitment due to ocean forcing, especially if the long-term retreat accesses more- or less-stable terminus positions.





**Table 2.** Parameters varied between three idealized glaciers (top) and the resulting steady-state values (bottom) calculated by the two-stage model.

| Parameter | Glacier 1 | Glacier 2 | Glacier 3 |
|---|---|---|---|
| Surface mass balance, $\bar{S}$ (m yr$^{-1}$) | 0.5 | 0.6 | 0.3 |
| Buttressing, $\Theta$ | 0.7 | 0.75 | 0.6 |
| Bed elev. at divide, $b_0$ (m) | $-100$ | $+150$ | $+100$ |
| Bed slope, $b_x$ | $-2 \times 10^{-3}$ | $-3 \times 10^{-3}$ | $-1 \times 10^{-3}$ |
| **Steady-state value** | | | |
| Interior thickness, $\bar{H}$ (m) | 1413 | 1569 | 2814 |
| Length, $\bar{L}$ (km) | 185 | 212 | 700 |
| Grounding line thickness, $h_g$ (m) | 526 | 545 | 673 |
| Fast response time, $\tau_F$ (yr) | 77 | 56 | 144 |
| Slow response time, $\tau_S$ (yr) | 2030 | 1160 | 4590 |

The slow mode also means that there can be large uncertainties in committed change if the magnitude of forcing is uncertain.

Figure 5d shows the response to a trend in $\Omega$, ranging from a 20–40% increase from the initial value. As of 2020, the differences in observed retreat are small, but diverge as the system approaches equilibrium over subsequent millennia. In other words, the slow response limits how "wrong" short-term simulations might be due to errors in the assumed forcing, but makes no such constraints on the committed change. The same principle also applies to uncertainty in the outlet glacier's length sensitivity: note that glaciers 1 and 2 have nearly identical responses on centennial timescales, but different equilibrium sensitivities.

Similar arguments have been made for uncertainty in radiative forcing and equilibrium climate sensitivity (Armour and Roe, 2011).

### 3.2 Emergence and detectability

We now turn to the topic of attributing outlet glacier retreats to natural or anthropogenic forcing. The attribution of an observed change to a particular cause can be a challenge because of factors specific to individual glaciers, such as complexities in

bed geometry, regional climate, or the local collection of ice-ocean interactions. It can also be a challenge because of factors intrinsic to the ice dynamics. We focus here on this latter set, and in particular on the contrasting effects of ocean vs. interior forcing.

Attribution is often framed in terms of the signal-to-noise ratio (SNR) of an observed trend. For a variable $x$, the SNR is often defined as $\Delta x / \sigma_x$, where $\Delta x$ is the change estimated by a linear fit and $\sigma_x$ is the standard deviation. For glacier retreat (or

changes in any analogous system), this depends on the SNR of the relevant forcing variable (e.g., surface temperature, ocean heat content, etc.), and also on the memory timescale over which the glacier integrates anomalies in that variable. A glacier's memory damps its response to high-frequency noise, but the tradeoff is that it also delays its response to a persistent trend.

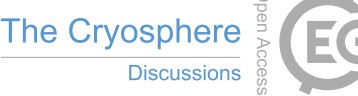

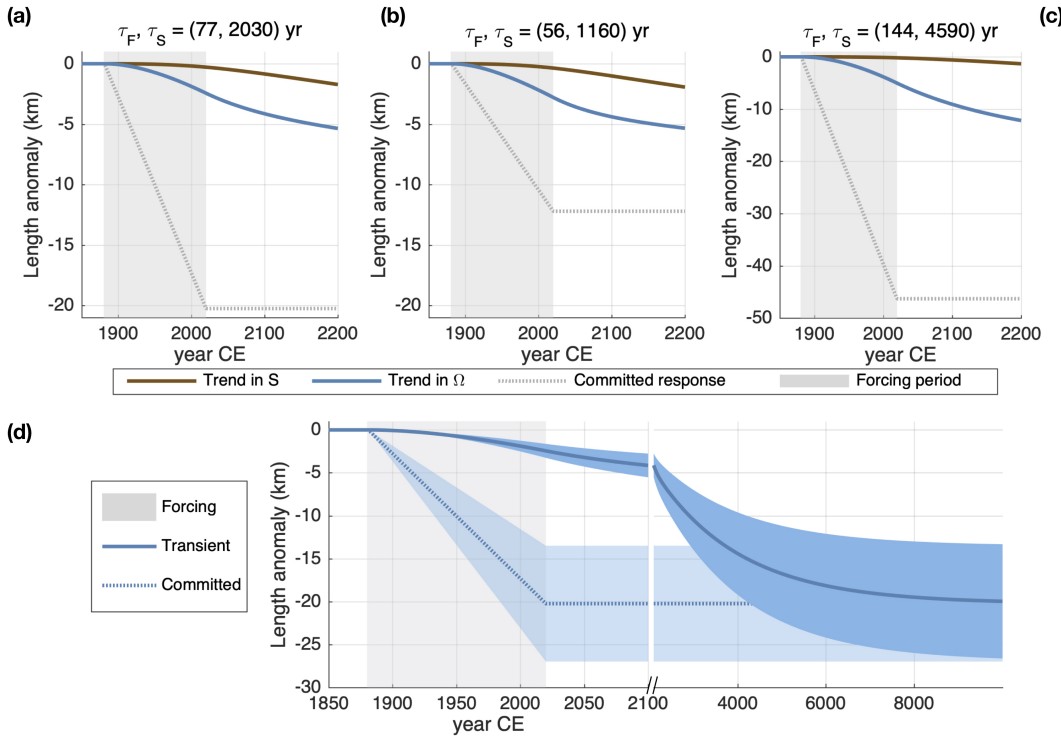

**Figure 5. (a)** Idealized climate forcing over the industrial era, for glacier 1 (see table 2). A 30% change in $S$ (brown) or $\Omega$ (blue) is realized as a linear trend from 1880–2020 CE. Dotted lines show the grounding-line position that would balance accumulation and grounding-line fluxes. **(b)** As for **(a)**, but for glacier 2, which has faster response times and smaller equilibrium sensitivity. **(c)** As for **(a)** and **(b)**, for glacier 3, which has slower response times and greater equilibrium sensitivity. **(d)** The range of responses of glacier 1 due to an uncertain forcing, idealized as a trend in $\Omega$ reaching a 20–40% change by 2020. The slow transient response means a small range in absolute length change in the modern era, but uncertainty in the forcing projects directly onto the committed equilibrium response. Note the break in the time-axis scale at 2100.

Roe et al. (2017) showed that mountain glaciers exemplify this concept and moreover, that their multidecadal response times are optimal for damping interannual climate variability while responding sensitively to centennial trends, thereby producing an
amplified SNR.

For outlet glaciers, however, two types of forcing and much longer response times are at play. To understand how these factors interact on the timescales of historical anthropogenic forcing, we consider three idealized scenarios of stationary variability plus a trend (Fig. 6a). Cases 1 and 2 have variability with no interannual persistence (i.e., white noise) in $S$ and $\Omega$, respectively. Case 3 has multidecadal variability in $\Omega$, generated by an AR-1 process with a memory timescale of 20 yr (see Sect. 2.7). To
compare glacier responses, we standardize the forcing scenarios by their SNR: In each case, a linear trend begins in 1880 CE (negative in $S$ for case 1, positive in $\Omega$ for 2 and 3). We stipulate that the trend reaches a 20% departure from the mean by





2020 CE, and continues thereafter at the same rate. The stationary variability again has $\sigma = 20\%$ of the initial mean. Thus, the imposed trends in climate forcing all have a SNR that reaches 1 by 2020.

Figure 6b shows the resulting grounding line anomalies of the test glacier described in section 2, generated with the two-
stage model. In all cases, the simulation was initialized in the year 0 CE. For reference, responses without variability are also shown (thin lines), as well as responses with variability but no trend. Shading indicates the $1\sigma$ and $2\sigma$ bounds of grounding-line fluctuations, determined from $10^7$ yr simulations of stationary variability.

In case 1, noisy surface mass balance drives small length fluctuations, but the very slow response to the trend in $S$ hampers detectability; the forced response is within $2\sigma_L$ until the late 21st century. In contrast, the faster response to a trend in $\Omega$
exceeds $5\sigma_L$ by 2020, for case 2 (interannual variability). The multidecadal fast response time thus acts to amplify the climate trend's SNR roughly by a factor of 5, consistent with the findings of Roe et al. (2017) for mountain glaciers. However, because persistence amplifies natural glacier variability (Sect. 2.7), the multidecadal variability in case 3 inflates the envelope of natural glacier fluctuations such that the forced response is again hard to detect on centennial timescales. The forced response is roughly $1\sigma_L$ by 2020, meaning that glacier dynamics no longer improve upon the SNR of the forcing. Thus, a multidecadal
fast timescale does not necessarily amplify the SNR of a climate trend, if a glacier is subjected to multidecadal ocean variability.

These idealized cases are useful for understanding factors that affect the detectability of a length trend, but they also demon-strate that the SNR may be a problematic metric in a practical sense. Because most of the natural glacier variability is expressed at very low frequencies, $\sigma_L$ (i.e., the noise) is undersampled by direct observations, which cover only several decades for most outlet glaciers. Additionally, slow natural excursions could dominate the baseline against which a trend is measured. For exam-
ple, in case 3, the *initial* length anomaly due to random variability is greater in magnitude in 1880 than the forced component of the retreat from 1880–2020 (i.e., the difference between orange and blue lines).

An alternative approach is to evaluate an observed $\Delta L$ against the distribution of natural trends that occur over the same interval of time as the observation. This incorporates information about the rate and persistence of natural length fluctuations. The simplicity of the two-stage model allows us to generate large synthetic ensembles of random fluctuations for a range of
glacier parameters and forcing scenarios. We develop distributions of naturally-forced trends as follows: If $\Delta t_{obs}$ is the length of a hypothetical observational period, we draw the last $\Delta t_{obs}$ years from model runs of $10^4$ years, each of which are forced with stationary noise and no trend. For each realization, $\Delta L_{null}$ is the trend in grounding-line position over the observational period. We find $\Delta L_{null}$ for $10^4$ independent simulations for each type of variability, yielding distributions of $\Delta L_{null}$ for a given $\Delta t_{obs}$. The resulting distributions are shown in Fig. 6c, for $\Delta t_{obs}$ of 50, 100, 250, and 500 years.

For variability in $S$, the distribution of $\Delta L_{null}$ is narrow for short $\Delta t_{obs}$ due to the strong damping of high frequencies, but widens as $\Delta t_{obs}$ increases. In other words, large fluctuations are possible, but ice dynamics constrain them to be slow. As expected, $\Omega$ variability increases the likelihood of observing larger trends on centennial timescales, and the effects of persistence are once again dramatic: a 1 km retreat in 50 years is very rare with no persistence (99th percentile), but fairly commonplace with multidecadal variability (70th percentile).

As $\Delta t_{obs}$ increases out to 500 years, the distributions of $\Delta L_{null}$ do not widen as much for variability in $\Omega$ as for variability in $S$. This suggests that centennial timescales (i.e., a few multiples of $\tau_F$) are nearly optimal for sampling large, persistent trends





driven by stochastic $\Omega$ variability alone (note that in the limit of $\Delta t_{obs} \gg \tau_S$ , the distributions would narrow back to zero). This raises the importance of the fast mode for attributing changes in the time frame of one-to-two centuries of anthropogenic climate forcing.

The widely-separated dynamical timescales characteristic of outlet glaciers pose a unique challenge for understanding their modern changes. The slow mode means that the overall variance ($\sigma_L^2$) must be considered, as stochastic variability over the last millennia may be important for the preindustrial state. Yet if the glacier is sensitive to variability at the terminus, the fast mode enables shorter-term fluctuations that may obscure the early response to anthropogenic forcing, especially if the variability has significant persistence. The sensitivity of grounding line flux to ocean variability and large calving events, and the associated

timescales, will thus be critical for attribution studies. Finally, it should be borne in mind that these detection challenges arise from slow dynamics, and not from low sensitivity. As is clear from Fig. 5, only a small fraction of the committed response is available for attribution studies today.

### 3.3   Inherited conditions

We have thus far considered outlet-glacier fluctuations due to stationary interannual-to-multidecadal climate variability. How-
ever, long response times also imply some memory of climate variations throughout the Holocene. Ice-sheet models are often "spun up" using paleoclimate proxy data precisely for this reason (e.g., Bindschadler et al., 2013). However, these strategies do not always reproduce the same modern ice extent (e.g., Goelzer et al., 2018), and the simulated ice-sheet history can depend strongly on the choice of proxy and its implementation in the model (e.g., Nielsen et al., 2018; Buizert et al., 2018). Here, we compare outlet glacier responses to ocean versus interior forcing over the Holocene as an additional factor for the modern
state. We focus on climate anomalies that are well-documented in the Northern Hemisphere, but similar considerations would apply to Antarctic outlet glaciers.

    First, we consider a climate scenario with idealized representations of three events: a deglacial warming at 11 kya, a Little Ice Age (LIA) cool period from 1450 to 1850 CE, and an anthropogenic warming trend from 1880 to 2000 CE. The deglacial and LIA transitions are smoothed by error functions of 500 yr and 50 yr widths, respectively. The magnitudes of deglacial,
LIA, and anthropogenic events have a ratio of 10:1:2. The scenario is initialized in equilibrium with the pre-Holocene climate. We scale forcings linearly to the climate anomalies, where warming corresponds to negative $S'$ and positive $\Omega'$. For $S'$, this implicitly assumes a temperate setting, but, again, we focus on the response timescales and not the surface mass balance of a particular setting. Additionally, a combination of forcings could be expected in reality, but these experiments serve as limiting cases to illustrate the relative influences of ocean and interior forcing over different timescales. This scenario is designed to
explore two practical points: (1) the glacier's memory of large, long-ago events compared to smaller, more recent events; and (2) the glacier's relative memory of past ocean vs. interior forcing.

    Figure 7a shows the idealized climate (top) and two-stage model responses for glacier 1 (bottom; $\tau_F, \tau_S \sim 76, 2000$ yrs). Anomalies are shown relative to the mid-Holocene and normalized to the large deglacial transition. Figure 7b shows 1000 to 2000 CE in more detail, including a scenario with no anthropogenic warming (dashed).



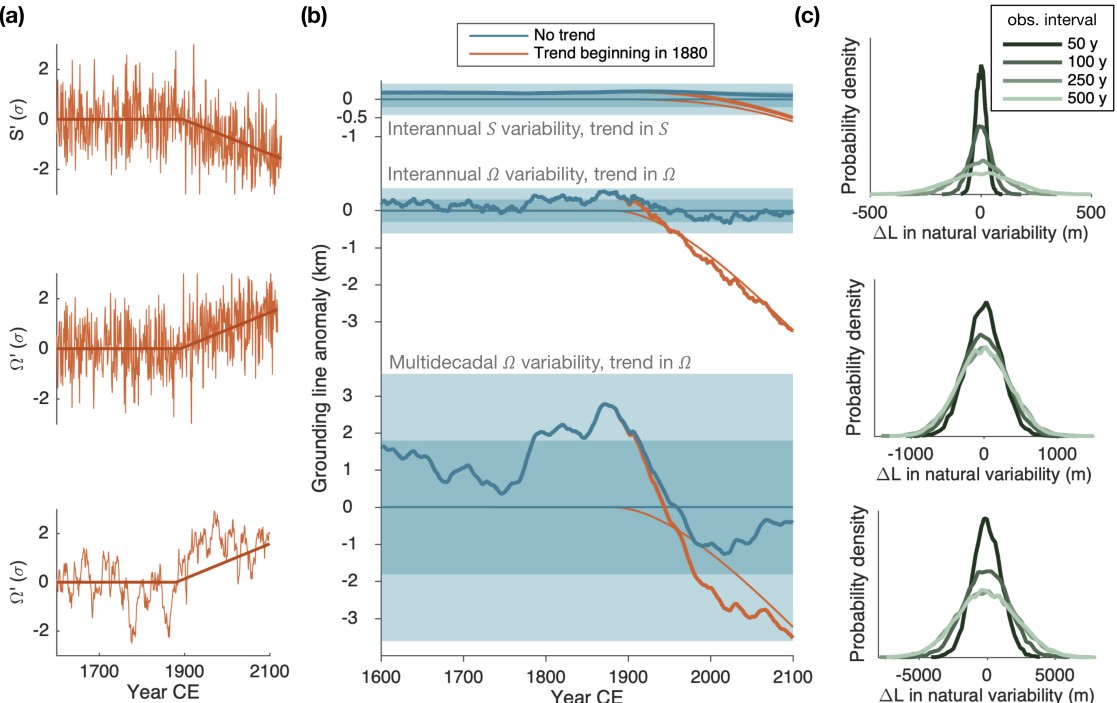

**Figure 6.** Detecting the response to a climate trend in the presence of natural variability. Three types of variability are considered in each panel: interannual variability in $S$ (top); interannual variability in $\Omega$ (middle); and multidecadal variability ($\tau_{AR1}$ = 20 yr) in $\Omega$ (bottom). **(a)** Idealized climate trends beginning in 1880. In all cases, the linear trend reaches a SNR of 1 by 2020. **(b)** Grounding-line responses to each idealized trend. Shaded regions are the $1\sigma_L$ and $2\sigma_L$ bounds for each type of noise. The orange lines indicate when the trend has been applied. Thinner lines show the grounding-line response without variability. **(c)** Probability density functions for grounding-line trends driven only by natural variability, over time periods from 50–500 years. Note the different length scales in each case. Trends on the order of km century$^{-1}$ are extremely unlikely to occur due to variability in $S$ alone (top), but commonplace if the glacier is sensitive to multidecadal variability in $\Omega$ (bottom).

For both forcings, the grounding-line retreat due to the deglacial signal is $99\%$ complete by the onset of the LIA. However, the advance following the LIA is $\sim 2\times$ greater for forcing in $\Omega$, because it can engage the fast mode to a greater degree. Yet, even forcing in $\Omega$ yields a muted response: the imposed LIA climate anomaly is $10\%$ of the magnitude of the deglaciation anomaly, yet the transient length response is only $\sim 3\%$ of the magnitude of the deglaciation response. The slow mode, which takes up the majority of the response for both forcing types, barely feels our 400 yr LIA before it is reversed. This is worth

bearing in mind whenever the duration of glacier excursions and climate anomalies are less than $\tau_S$, because the system never achieves equilibrium. This would be an issue particularly if such events are used to tune glacier sensitivity in models.

We also consider the response to a more realistic forcing timeseries. Obviously, this is not intended as an actual reconstruction of terminus changes, but it is useful to see how glacier memory integrates the continuum of variations found in paleoclimate



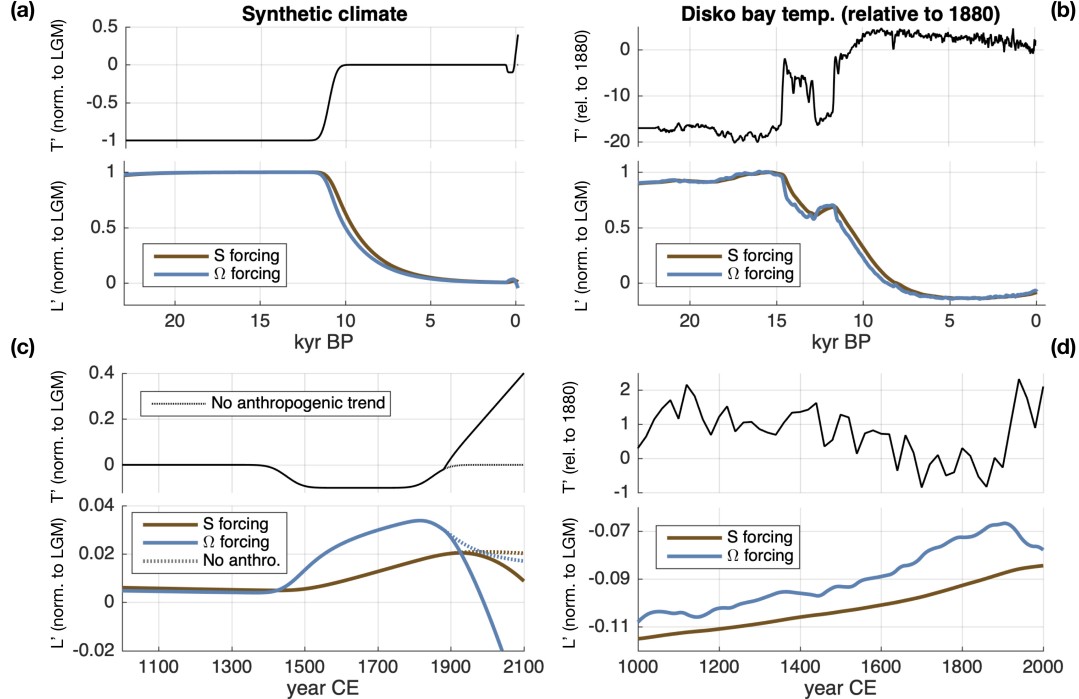

**Figure 7.** Response to long-term climate variations. **(a)** An idealized climate scenario representing the last deglaciation, Little Ice Age (LIA), and anthropogenic forcing (black line). Normalized length responses of the linear model are shown for forcing applied via $S$ (brown) and $\Omega$ (blue). **(b)** As for **(a)**, but zoomed into the last 1000 years. $\Omega$ forcing yields a much larger response to the synthetic LIA and anthropogenic trend. **(c)** A more realistic forcing timeseries, based on reconstructed atmospheric temperatures for Disko Bay (black line; Buizert et al., 2018). Normalized length responses are again shown for both forcing types, which yield similar responses on multimillennial timescales. **(d)** As for **(c)**, but zoomed into the last 1000 years. Again, the LIA is expressed to a greater degree through terminus forcing. Note also the long-term advance driven by gradual cooling over the last several millennia.

records. We use a timeseries of temperatures for Disko Bay, Greenland, from the regional reconstruction of Buizert et al. (2018)

(Fig. 7c–d). The forcings are scaled as above, and normalized to 1880 CE.

The glacier responses in Fig.7d show the same essential behavior as the more idealized case in Fig. 7b. Even with a more subtle LIA, the terminus response is much more pronounced if it is driven by $\Omega$ anomalies that engage the fast response. It is worth noting, though, that most of the glacier disequilibrium in the 1800s is due to cooling over the previous millennia, to which the slow mode responds with a pronounced lag. Thus, the preindustrial state of an outlet glacier may depend significantly

both on LIA and longer-term ocean cooling, via its two distinct response timescales.





## 4   Summary and discussion

Marine-terminating outlet glaciers are sensitive to two fundamental types of forcing: changes in surface mass balance, which is distributed over a large interior catchment; and changes in their flux at the grounding line, which are typically driven by the ocean. We have used two models of different complexity to explore and contrast the dynamic responses to these two categories

of forcing. Our key findings are as follows:

1.   Ocean forcing (via the grounding-line flux coefficient, $\Omega$) and interior surface-mass-balance forcing elicit fundamentally different transient grounding-line responses (Figs. 1 and 2). The response to ocean forcing is characterized by a fast initial grounding-line migration, followed by a second, much slower stage of adjustment. In contrast, the response to interior forcing is dominated by slow grounding-line migration, with very strong damping at short timescales.

2.   The two-stage model (Robel et al., 2018) captures the evolution of flux anomalies that gives rise to these two contrasting dynamic responses (Fig. 3). This lends further confidence to the two-mode interpretation of outlet glacier dynamics: a fast mode associated with adjustment of the grounding zone, and a slow mode associated with the interior reservoir. We have shown here that grounding-line forcing projects onto both modes, while interior forcing projects almost entirely onto the slow mode.

3.   Persistence in stochastic climate forcing amplifies natural grounding-line fluctuations (Fig. 4). Increased persistence means more of the variance in the forcing occurs at low frequencies, where the glacier is ultimately more sensitive. In particular, multidecadal ocean variability can drive large fluctuations on multidecadal to millennial timescales. Understanding the magnitude and persistence of ocean forcing is thus critical for attributing observed terminus changes and detecting the response to anthropogenic forcing (Fig. 6).

4.   Despite contrasting responses on short timescales, slow dynamics dominate the long-term response for both types of forcing. This implies a large committed response (Fig. 5), as well as a memory of past climate fluctuations (Fig. 7), for both types of forcing. The slow response of interior ice and the attendant consequences have been explored in a number of previous studies (e.g., Nye, 1960; Levermann et al., 2013; Robel et al., 2018); the results presented here demonstrate how a slow response is fundamental to ocean forcing as well.

Given the rapid observed changes linked to ocean forcing in the past few decades, it is useful to discuss points (3) and (4) further. Increases in discharge from Greenland outlet glaciers have been linked to regional climate variability and warming of the subpolar North Atlantic (Straneo and Heimbach, 2013). Records for a smaller selection of glaciers extend up to a century (Andresen et al., 2012; Bjørk et al., 2012) and futher indicate sensitivity to regional oceanic forcing on relatively short timescales. Evidence for the response to decadal variability is strong in Antarctica, too (e.g., Jenkins et al., 2016, 2018). In sum,

the capacity for outlet-glacier grounding lines to react quickly to changes at the terminus is quite clear from the observational record.

Our results show that climate and glacier fluctuations over the historical record also have a longer-timescale context. The widely-separated glacier response timescales mean that fast fluctuations are essentially superimposed on millennial fluctua-





tions. This is clear even in the flowline model, which does not make an explicit approximation of only two timescales (e.g.,
Fig. 1d). Slow fluctuations are difficult to resolve in observational records, but the practical point is that short periods of ter-
minus "stabilization", or fluctuations about a *short-term* mean state do not necessarily preclude a large disequilibrium between
fluxes near the grounding line and fluxes from the interior. This disequilibrium should be reflected in ice-thickness and velocity
profiles, and has been noted for some systems whose retreat or discharge rates have leveled off, including Jakobshavn Isbrae
(e.g., Joughin et al., 2012b; Khazendar et al., 2019) and Pine Island Glacier (e.g., Christianson et al., 2016). Careful modeling
might integrate such observations with climate reconstructions to inform attribution studies or model initialization. Wherever
the forcing history is uncertain, multiple realizations of past variability should be considered, and we again emphasize that the
persistence of this variability is a critical parameter.

A number of simplifying assumptions throughout our study warrant some discussion. First of all, we have assumed a con-
stant, prograde bed slope. In reality, variations in bed topography can have a strong effect on retreat rates and sensitivity, both at
the terminus (Catania et al., 2018) and in the interior (Felikson et al., 2017). The effect of these variations on the fast and slow
modes remains to be investigated. Felikson et al. (2017) showed that bedrock knick-points may serve as barriers to rapid inland
thinning following forcing at the terminus. Depending on where the knickpoint is, this could mean that the dimensions of the
effective interior reservoir is different for ocean vs. interior forcing, potentially changing the relevant response timescales.

A related issue is the instability associated with retrograde slopes (e.g., Weertman, 1974; Schoof, 2007). In an unstable
configuration, the linearized timescales diverge, but RRH showed that fast and slow tendencies still govern the rates of unstable
retreat. More recently, Robel et al. (2019) showed that instability magnifies variability in transient retreat rate due to internal
climate variability. Together, these points suggest that instability might also magnify the difference between retreats initiated
by interior and grounding-line flux anomalies.

Another simplification is that we have focused on spatially-uniform interior forcings, whereas in reality, surface-mass-
balance anomalies are likely to be greater near the marine margin. We conducted several experiments with the flowline model
in which $S$ anomalies were concentrated towards the grounding line, and also amplified to produce the same volume anomalies.
This enhanced the high-frequency length response, although there was still a clear difference from the transient response to
$\Omega$ forcing. Compared to a uniform $S$ anomaly, a step change in $S$ concentrated on the lower half of the glacier (with doubled
magnitude) nearly doubles the grounding-line response after 100 years, while the 100 yr response following a step in $\Omega$ is four
times greater. For white-noise forcing, $\sigma_L$ increases $\sim 12\%$ if $S$ anomalies are concentrated in the same way, but increases
$> 50\%$ if applied as $\Omega$ anomalies. Thus, the temporal distinction between interior and ocean forcing will depend on the spatial
pattern of mass balance, but it would likely take extremely large and concentrated anomalies to match the fast response of $\Omega$
forcing. We expect that the distinction would also depend on the horizontal catchment geometry: convergence would amplify
the effects of anomalies from the deep interior, even if they are smaller in magnitude. Further experiments with 2-D geometries
would help to clarify these potentially competing effects. We have also neglected orographic and surface-elevation feedbacks,
which would depend on the spatial pattern of thinning, and may thus affect fast and slow responses differently. These would
provide another interesting avenue for future analyses.





Finally, we chose to impose ocean forcing via the grounding-line-flux coefficient, which adjusts the parameterized relationship between ice thickness and grounding-line flux. This allowed us to compare flux perturbations in a very general way, but it

would be more realistic to directly force a dynamic ice shelf or to perturb calving processes. The analytical flux conditions of Haseloff and Sergienko (2018) could be an intermediate route, and would be feasible for models that parameterize grounding-line flux. RRH showed that these conditions can introduce nonlinear sensitivity to perturbations in a buttressing ice shelf, and thus another key difference between glacier responses to interior and ocean forcing. For example, this nonlinearity would skew the distribution of fluctuations driven by natural ocean variability, changing the thresholds for trend detection (i.e., Fig. 6).

Previous studies have employed a variety of forcing strategies for glaciers without buttressing ice shelves. Some have directly perturbed stresses at the grounding line (e.g., Nick et al., 2009; Price et al., 2011), while others have implemented forcing via frontal melt rates (e.g., Morlighem et al., 2016; Aschwanden et al., 2019). Another approach is to impose terminus retreat based on observations, and then allow up-glacier ice thickness and flux to evolve. This approach has been used to isolate the effects bed topography on inland thinning (Felikson et al., 2018).

Regardless of how forcing is implemented, our analyses show that the coupling between grounding-zone and interior dynamics is a key part of an outlet glacier's response to ocean forcing. In particular, increased discharge is what eventually precipitates the slow (but large) second stage of grounding-line retreat. Our idealized grounding-line forcing has limitations, of course, but the mechanism for a second stage of retreat is physically robust: elevated interior fluxes must eventually fall as the interior drains, creating a tendency towards further grounding line retreat (Fig. 3). The actual retreat could be modulated depending on

the bed topography through which the grounding line migrates, and this would be yet another factor to investigate with more realistic geometries.

At the ice-sheet scale, ocean forcing must often be simplified considerably, and the optimal strategy remains to be determined. The Ice Sheet Model Intercomparison Project for CMIP6 (ISMIP6; Nowicki et al., 2016) defines several experimental protocols for use across various ice sheet models. The ISMIP6 ocean-forcing parameterizations for Greenland were recently

established, based on an empirical study of a large collection of terminus observations and regional climate data (Slater et al., 2019a, b). They proposed two strategies: one in which length change is imposed directly as a function of climate forcing ("retreat implementation"), and one in which submarine melt rates are prescribed ("melt implementation"). Both leave the evolution of ice flux up to the ice sheet model, but the retreat implementation does not allow interior ice dynamics to feed back into the terminus position. The results presented here suggest that, because the timescale of interior dynamics is much longer

than the observational records used to calibrate the imposed retreats, the retreat implementation would project less long-term terminus retreat compared with the melt implementation.

The impact that the choice of parameterization would have on projections of near-term sea level rise is not immediately clear, however. Slater et al. (2019a) note that these parameterizations are intended primarily for 21st century projections. In a scenario with severe warming, the fast response to progressive forcing might plausibly outweigh the contribution of the slow

mode over this period. On the other hand, in a scenario of climate stabilization, the two approaches could yield more disparate results. Under the retreat implementation, the terminus stabilizes if the climate stabilizes (Slater et al., 2019b), whereas under the melt implementation, it could in principle continue retreating due to disequilibrium between interior and grounding-zone





fluxes (much like in Fig. 5). In such a case, the choice of parameterization would affect the difference between responses to low versus high emissions scenarios (e.g., RCP 2.6 vs. 8.5). The principles from our idealized experiments suggest that a fixed

versus free terminus would yield a different partition between fast and slow glacier responses, but further experimentation is needed to investigate this. Although much attention is directed toward sea-level rise by 2100, longer-timescale comparisons within in the ISMIP6 framework, both with idealized and comprehensive models, may also prove illuminating.

Understanding the timescales of glacier dynamics is a long-standing pursuit in glaciology (e.g., Nye, 1960; Jóhannesson et al., 1989; Oerlemans, 2001; Roe and Baker, 2014; Robel et al., 2018). Here, we have explored for marine-terminating

outlet glaciers how these timescales manifest under different types of forcing. Interpreting observations and making useful projections will always depend partly on characteristics that are unique to each outlet glacier, and on processes that remain to be investigated further. However, the basic constraints of flux conservation and large-scale geometry will always apply, and, as we have shown here, are enough on their own to yield consequential differences in glacier response. Obviously, there are many intermediate levels of complexity between the models we have used here and those used for detailed simulations and

projections. Alongside the need for more-comprehensive glacier and ice-sheet projections in a warming climate, there also remains the need to improve understanding by analyzing model experiments positioned throughout this hierarchy of model complexity (e.g., Held, 2005). Reduced models and experiments with idealized glacier geometries, in which the fundamentals are laid bare, can thus serve as a useful foundation for the needed analyses.

*Code and data availability.* Code used for the analyses in this study will be made available as a public repository at https://github.com/johnerich/outlet-

glac-forcing. Output from the flowline model is available from the corresponding author upon request.

## Appendix A: Linearized two-stage model

Here we present the linearized two-stage model in more detail, and derive a discrete autoregressive form. Again, a full model derivation can be found in RRH; we start here from the linearized equations with perturbations in $Q_g$ and $S$:

$$\frac{\partial H'}{\partial t} = A_H H' + A_L L' + \frac{1}{\bar{L}}\left(\frac{\bar{H}}{\bar{h}_g} - 1\right)Q_g' + S' \tag{A1}$$

$$\frac{\partial L'}{\partial t} = B_H H' + B_L L' - \frac{1}{\bar{h}_g}Q_g'. \tag{A2}$$

$A_H$, $A_L$, $B_H$, and $B_L$ are the linear couplings between length and thickness anomalies:

$$A_H(\bar{H}, \bar{L}) = -\bar{Q}_g \alpha(\bar{h}_g \bar{L})^{-1} \tag{A3}$$

$$A_L(\bar{H}, \bar{L}) = \bar{Q}_g \bar{L}^{-2}\left[1 + \gamma\bar{H}\bar{h}_g^{-1} + \beta\lambda b_x\bar{L}\bar{h}_g^{-1}(1 - \bar{H}\bar{h}_g^{-1})\right] \tag{A4}$$

$$B_H(\bar{H}, \bar{L}) = \bar{Q}_g \alpha(\bar{H}\bar{h}_g)^{-1} \tag{A5}$$

$$B_L(\bar{H}, \bar{L}) = \bar{Q}_g \bar{h}_g^{-1}(\beta\lambda b_x\bar{h}_g^{-1} - \gamma\bar{L}^{-1}), \tag{A6}$$





where $\lambda = \rho_w/\rho_i$. For numerical implementation, Eqs. (A1) and (A2) can be discretized using a backward-Euler method, where $\Delta t$ is the timestep between glacier states $[i]$ and $[i-1]$:

$$\frac{H'_{[i]} - H'_{[i-1]}}{\Delta t} = A_H H'_{[i]} + A_L L'_{[i]} + \frac{1}{\bar{L}}\left(\frac{\bar{H}}{\bar{h}_g} - 1\right)Q'_{g[i]} + S'_{[i]} \tag{A7}$$

$$\frac{L'_{[i]} - L'_{[i-1]}}{\Delta t} = B_H H'_{[i]} + B_L L'_{[i]} - \frac{1}{\bar{h}_g}Q'_{g[i]}. \tag{A8}$$

Solving for $H_{[i]}$ and $L_{[i]}$ gives:

$$H'_{[i]} = \frac{1}{1 - A_H \Delta t}\left[A_L \Delta t L'_{[i]} + H'_{[i-1]} + \frac{1}{\bar{L}}\left(\frac{\bar{H}}{\bar{h}_g} - 1\right)\Delta t Q'_{g[i]} + \Delta t S'_{[i]}\right] \tag{A9}$$

$$L'_{[i]} = \frac{1}{1 - B_L \Delta t}\left[B_H \Delta t H'_{[i]} + L'_{[i-1]} + \frac{1}{\bar{h}_g}\Delta t Q'_{g[i]}\right], \tag{A10}$$

which are still semi-implicit in terms of the current state $[i]$. However, we can substitute Eq. (A10) for the $L'_{[i]}$ term in Eq. (A9), and vice versa for $H_{[i]}$. Solving again for $H_{[i]}$ and $L_{[i]}$, we arrive at explicit expressions that depend on only the past state

and current forcing. This takes the form of a two-dimensional autoregressive process (e.g., Box et al., 2015) with two forcing terms. For compactness, we redefine several parameter combinations at this point. Let $\epsilon = (1 - B_L \Delta t)^{-1}$, $\eta = (1 - A_H \Delta t)^{-1}$, and $\kappa = (1 - \eta\epsilon A_L B_H \Delta t^2)^{-1}$. Then,

$$\begin{bmatrix} H'_{[i]} \\ L'_{[i]} \end{bmatrix} = \mathbf{C}\begin{bmatrix} H'_{[i-1]} \\ L'_{[i-1]} \end{bmatrix} + \boldsymbol{d}Q'_{g[i]} + \boldsymbol{e}S'_{[i]} \tag{A11}$$

where the autoregressive coefficients are contained in the matrix $\mathbf{C}$ and vectors $\boldsymbol{d}$ and $\boldsymbol{e}$:

$$\mathbf{C} = \kappa\begin{bmatrix} \eta & \epsilon\eta A_L \Delta t \\ \epsilon\eta B_H \Delta t & \epsilon \end{bmatrix} \tag{A12}$$

$$\boldsymbol{d} = \kappa\Delta t\begin{bmatrix} \eta\left(\bar{L}^{-1}(\bar{H}/\bar{h}_g - 1) - \epsilon A_L h_g^{-1}\Delta t\right) \\ \epsilon\left(B_H \bar{L}^{-1}(\bar{H}/\bar{h}_g - 1)\Delta t - \bar{h}_g^{-1}\right) \end{bmatrix} \tag{A13}$$

$$\boldsymbol{e} = \kappa\eta\Delta t\begin{bmatrix} 1 \\ \epsilon B_H \Delta t \end{bmatrix}. \tag{A14}$$

$$\tag{A15}$$

This form lends iteslf to a straightforward solution algorithm, given steady-state glacier parameters and forcing timeseries as

inputs. Note that for the grounding-line forcing used in this study, $Q'_g$ is proportional to $\Omega'$, but other relationships could easily be implemented.

*Author contributions.* All authors contributed to the study design. JEC carried out the analyses and wrote the manuscript, with input from all authors.



*Competing interests.* The authors declare no competing interests.

*Acknowledgements.* We are very grateful to David Pollard (Penn State Univ.) for model code and a generous introduction to using the flowline model. We also thank Ginny Catania, Fiamma Straneo, and Donald Slater for insightful conversations on outlet glacier dynamics. JEC was supported by the NSF Graduate Research Fellowship Program (DGE-1256082).



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
