# Peer review of "The contrasting response of outlet glaciers to interior and ocean forcing"

_The Cryosphere, 2019_

## Referee Comment (RC1) · Martin Lüthi (Referee) · 11 Apr 2020

Dear Colleagues

This is a very nice manuscript, well presented, with a good methodology and a thorough set of model experiments that comes to significant and interesting conclusions.

There are only a few minor things I would like to see changed, and which are indicated below.

Best regards,

Martin Lüthi

==================

[Figure]

General comments:

- clearly written

- flotation-based calving criterion -> what would change with other parametrizations?

- call "interior forcing" something more like "mass balance forcing"

- The paper structure, even if it is not following the standard pattern, is useful and helps guiding the reader through the manuscript.

- The bibliography should be carefully revised. Capitalization of journal names is often wrong, and DOIs are missing. IPCC (in Stocker..) should be mentioned, etc..

Style (to be adapted to journal standard everywhere):

- do not put variables in parentheses (as is occasionally done)

- no colons before equations

- write out "Equation" in the text, only abbreviate in parantheses using "Eq." oder "Sect." is not commonly used in manuscripts.

- use real fractions in equation environments:

\frac{1}{2} etc

Specific comments:

76 variable names should consistently *not* be enclosed in parentheses. Also indicated that this is just a statement of conversation of mass.

86 "Glen-type coefficient": better say you use power-law rheology with A and n.

89 In an equation you should use real fractions: \frac{1}{2} also in Eq. (7) etc \frac{\tau_b}{C}^{\frac{1}{m}} and so on,

please change everywhere in the manuscript

105 so, here you allude that your model is defined on a grid?

115 "Section" (not abbreviated in the text)

- Figure 1d: it would be interesting to also display the forcing (maybe as gray line with appropriate scaling).

- Table 1: why is the sliding coefficient given with 5 significant digits? I think in this study this can be any arbitrary number of about that order of magnitude. Same for A.

Scientific notation does not use a \cross, but a \cdot

Also consider two more columns, one for the symbol (should come first) and one for the units.

Table header should not be given in bold.

164 it is not clear what the "two stages" are. Some other designation might better describe the model.

165 "static geometry" (as opposed to "dynamic") is probably what is assumed by all models. Do you mean "linear/sloped"?

167 "S\cross L" should be "S\cdot L" or just "SL". Do not put variable names in parentheses in some places.

167 "Q" : what is this, the flux into the grounding zone? Please make description clearer.

190 Repeat which variable is the "mass balance rate" (I think S)

192 What is this "small reservoir"? Until now there was just one, described by S and L. Maybe this is the purple box in Figure 2. If so, this setup should be made more explicit from the beginning, and Figure 2 should be improved.

200 It is not clear what the use of a linearized system is. It is certainly useful to find eigenvalues and eigenvectors at a certain state.

But having a "full" model with all nonlinearities, and then using a linearized "deviation model" makes no sense for the large changes investigated. Also, the "full" model is extremely cheap to calculate, as it is just a 2-variable ODE.

Especially given the comments in line 210 that the nonlinear ("full") ODE give the same results makes one wonder, why the linearized version is used at all (except for investigation of eigenvalues).

250 Sn important additional argument in this discussion is also the spatial scale. Interior changes take a certain time, given by the ice flow speed, to affect the terminus, while processes at the terminus are immediately affecting glacier length. This is the same as on any glacier, but the effect of terminus dynamics in tidewater are much faster and bigger than those of terminus melt on a mountain glacier.

265 Here a link to kinematic wave speed would be very interesting. How fast is the terminus signal propagating upstream.

Figure 3: The two blueish-greenish colors are difficult to distinguish, use a better color table.

Figure 4: Use clearer colors, full red, blue etc. They might look less appealing, but are easier discernible.

308 "white noise forcing" (also 311)

346 where do these exact numbers come from?

350 should the "as of 2020" follow "transient response"?

Figure 5: caption: "Table" (upper case)

457 Here the question posed above gets more pressing: why is the linearized model used? Can it be used at all for such large changes? How wrong do the results get? Why is not the "full" simple model used here?

458 And then also: are the time scales always the same, even if the glacier geometry changes by a very large amount?

467 "time series" (two words)

---

## Referee Comment (RC2) · Anonymous Referee #2 · 18 Apr 2020

In this study, the authors used simplified models to explore the response of ice flow to different sources of climate forcing, including perturbations at the grounding line and to the surface mass balance in the ice interior. I found this approach to be novel and very interesting. It successfully provides new insights into the connections between various forcings, in terms of amplitude and effects operating at different timescales.

The manuscript is well organised and very well written, overall. I support its publication after minor corrections.

Comments: -L. 93: delete "model" in "The PD12 model model. . ."

-L. 203: It isn't clear to me, what does "a different flowline model" refer to. Is it simply the flowline model described in section 2.2?

[Figure]

-L. 339: correct "instantaneous equilibrium"

-L. 372, and first paragraph after subtitle: it would be helpful to define "emergence and detectability" more explicitly. As currently presented, I am not sure how the paragraph (lines 373-377) introduces the section.

-L. 381 and 382: Can you clarify how the glaciers' memory mentioned relates to the committed change discussed previously?

-L. 386: Specify which "two types of forcing" you are talking about (ocean vs interior, presumably?)

-L. 399: can you specify "detectability..." of what?

-L. 458: Did you mean a reference to Figure 7c?

Generally, I find the figures to be clear, although I would suggest working a bit more on / completing some of the figure legends, in particular for Figure 1, Figure 2, Figure 5 and Figure 6. For example:

Figure 1: insert "(blue)" and "(orange)" after "omega" and "S" respectively, in caption d.

Figure 2: same thing for caption c, d, and e.

Figure 5: Caption needs to be more precise: aren't panels a, b and c showing the response of glaciers to idealized climate forcing? As it stands, it reads as if they show the climate forcing itself.

Figure 6: Similarly, some details and descriptions are missing. E.g., suggest completing the legend for panel b. I would also suggest making the titles currently in grey stand out a bit more.

---

## Author Comment (AC1) · 16 May 2020

The comment was uploaded in the form of a supplement:
https://www.the-cryosphere-discuss.net/tc-2019-301/tc-2019-301-AC1-supplement.pdf

---

## Author Comment (AC2) · 16 May 2020

The comment was uploaded in the form of a supplement:
https://www.the-cryosphere-discuss.net/tc-2019-301/tc-2019-301-AC2-supplement.pdf
* * *

---

## Author Response (AR1)

*Dear Colleagues*

*This is a very nice manuscript, well presented, with a good methodology and a thorough set of model experiments that comes to significant and interesting conclusions.*

*There are only a few minor things I would like to see changed, and which are indicated below.*

*Best regards,*
*Martin Lüthi*

We thank Dr. Lüthi for the encouraging review, suggestions for clarification, and several very thought-provoking questions. Please find our responses and changes below.

*General comments:*

*- clearly written*

*- flotation-based calving criterion -> what would change with other parametrizations?*

Thanks for raising this question. The topic of assumptions at the grounding line (or calving front) is an important one. We are assuming the comment refers specifically to a case with no ice shelf (i.e., ice calves immediately at the grounding line) and thus no buttressing. In such a case the analytical form of $\Omega$ from Schoof (2007) would not be strictly valid. Many mechanisms might be implicated in ocean forcing in such cases, but given that reduced back-stress is often part of the picture (e.g., Nick et al., 2009; Joughin et al. 2012), we think forcing via the grounding line flux coefficient is likely to still be a good starting point for representing ocean forcing. As long as flux follows the general form of equation 1 ($Q_g = \Omega h_g^\beta$), we expect the main findings of the study to be applicable, at least in a qualitative sense.

Similar, but more detailed approaches to forcing have been considered previously. Nick et al. (2009) simulated several large outlet glaciers, perturbing the force balance and allowing the upstream glacier to evolve, but these simulations were limited to decadal timescales. We also discuss the similar experiments of Price et al. (2011) and Goldberg et al. (2015) in section 3.1

(committed change). These prior experiments have showed similar behavior to what we consider the "fast" dynamics, albeit with much more detail on short timescales. Extending this type of perturbation experiment to much longer timescales would be an interesting next step for investigating (a) how our results would change with more detailed dynamics at the glacier front, and (b) how their results extent to the eventual "slow" response.

We also discuss alternative grounding line flux rules in the discussion section, and reference the findings of Robel Roe and Haseloff ("RRH"; 2018), who explored additional nonlinearities associated with alternative solutions (see Hasleoff and Sergienko, 2018). The simplified flux condition and forcing scheme we used offered a very general approach for initial analyses, but we expect that considering alternative assumptions and dynamics (including calving) would yield further insights.

*- call "interior forcing" something more like "mass balance forcing"*

If the reviewers and editor find it acceptable, we would prefer to retain "interior forcing" throughout, we believe it helps highlight the geometric differences between interior mass balance and ocean forcing. That is, it may help remind the reader that it is distributed over the entire interior. (This of course is a simplifying assumption, and is addressed in the discussion.)

However, we agree that clarity in the language here is important. We have added a statement in section 2.3 (where the first forcing experiment is introduced) to establish that these terms will consistently refer to forcing described as such. Hopefully this makes usage of the term "interior forcing" clearer throughout.

In all model experiments throughout this study, "interior forcing" and "ocean forcing" will refer to perturbations applied in the following manner. Interior surface-mass-balance anomalies are assumed to be spatially uniform. We represent ocean forcing very simply by perturbing the grounding-line-flux coefficient, $\Omega$ …

In addition to this statement, we have made edits to ensure that usage is consistent throughout the manuscript. For example, there were some places where "terminus forcing" was used instead of "ocean forcing". Now, "interior forcing" and "ocean forcing" are used consistently when referring to our own experiments, or the parameters S and $\Omega$ are used as descriptors.

*- The paper structure, even if it is not following the standard pattern, is useful and helps guiding the reader through the manuscript.*

*- The bibliography should be carefully revised. Capitalization of journal names is often wrong, and DOIs are missing. IPCC (in Stocker..) should be mentioned, etc..*

The bibliography has been revised.

*Style (to be adapted to journal standard everywhere):*

We have tried to make the changes as suggested, but find some cases where colons and parentheses are necessary for clarity and/or sentence structure, and these cases seem to be consistent with usage in recent papers in *The Cryosphere*. However, we will defer to the editors on the preferred style.

*- do not put variables in parentheses (as is occasionally done)*

Changed throughout, except where variables are presented essentially as parenthetical statements (e.g., reminders).

*- no colons before equations*

We have removed most colons; however, there are some cases where it seems needed based on the sentence structure.

*- write out "Equation" in the text, only abbreviate in parantheses using "Eq." oder "Sect." is not commonly used in manuscripts.*

Fixed throughout.

*- use real fractions in equation environments:*

*\frac{1}{2} etc*

Fixed.

*Specific comments:*

*76 variable names should consistently \*not\* be enclosed in parentheses. Also indicated that this is just a statement of conversation of mass.*

Fixed. The conservation of mass has been made more explicit:

The evolution of local ice thickness h at a grid point reflects the balance of mass exchange at the surface and horizontal ice-flux divergence. Conservation of mass requires that

$$\frac{\partial h}{\partial t} = S - \frac{\partial(\bar{u}h)}{\partial x},$$

where…

*86 "Glen-type coefficient": better say you use power-law rheology with A and n.*

Fixed:

We assume a typical power-law rheology, with coefficient A and exponent n (e.g., Glen, 1955).

*89 In an equation you should use real fractions: \frac{1}{2} also in Eq. (7) etc*

*\frac{\tau_b}{C}^{\frac{1}{m}} and so on, please change everywhere in the manuscript*

Fixed.

*105 so, here you allude that your model is defined on a grid?*

The grid applies only to the flowline model, and not the two stage model. We have revised the text to clarify:

The PD12 model calculates grounding line flux based on a thickness hg that is linearly interpolated from the height above flotation of the last-grounded and first-floating grid cells, to the point where flotation is reached. The corresponding sub-grid grounding-line position is shown for all output from the PD12 model in this study. Although the PD12 model is typically run on coarse grids (O ~1–10 km) for continent-scale simulations over many millennia, we use a grid of 100 m to better resolve the details of grounding line variations.

*115 "Section" (not abbreviated in the text)*

Fixed.

*- Figure 1d: it would be interesting to also display the forcing (maybe as gray line with appropriate scaling).*

Good suggestion – we have added the noisy forcing to Fig. 1d.

*- Table 1: why is the sliding coefficient given with 5 significant digits? I think in this study this can be any arbitrary number of about that order of magnitude. Same for A.*

This is a good point – the exact value of C and A are not important to that level of precision. A and C affect the equilibrium grounding line position identically for both models (via \Omega). The treatment of interior fluxes is different for the two models, but provided the two-stage model captures interior thickness reasonably, it should emulate the flowline model over a range of values for A and C.

These values were initially chosen for comparison with results from Robel et al., 2018 (and are "default" values in several other studies). We would elect to keep them defined to full precision in the table for the sake of clarity, but would defer to the Editors on this.

*Scientific notation does not use a \cross, but a \cdot*

We will defer to editors on this … recent papers in *The Cryosphere* appear to use a cross (\times).

*Also consider two more columns, one for the symbol (should come first) and one for the units.*

Good suggestion. Columns added here, and for Table 2 as well.

*Table header should not be given in bold.*

Fixed.

*164 it is not clear what the "two stages" are. Some other designation might better describe the model.*

This is a good point – it was not initially clear as described. We would prefer to keep this terminology following Robel et al. (2018) (where the model is derived), and to maintain a connection to other "stage" models (Roe and Baker, 2014). However, we agree that this designation should be made clearer. We shifted introduction of the term to a later paragraph, when the two equations are presented. We then refer the reader to the later section (2.6), where we discuss the stages in more detail.

Two coupled equations capture the transient adjustment of the two degrees of freedom, $H$ and $L$ as they relax towards a steady state that balances all three fluxes:

$$\frac{\partial H}{\partial t} = S - \frac{Q}{L} - \frac{H}{h_g L}(Q - Q_g)$$

$$\frac{\partial L}{\partial t} = \frac{1}{h_g}(Q - Q_g).$$

Because achieving steady state requires adjustment of both $H$ and $L$ we refer to this model as the ``two-stage model", following RRH. We discuss the operation of these stages further in section 2.6…

And In section 2.6, we also added a link to the two-stage terminology to make this concept clearer:

These modes can also be conceptualized as a two-stage low-pass filter on any forcing time series…

*165 "static geometry" (as opposed to "dynamic") is probably what is assumed by all models. Do you mean "linear/sloped"?*

We have re-worded here to clarify:

The geometry is further described by a bed topography with constant average slope $b_x$ .

We also clarified description of the bed for the flowline model, noting that it is constant in time. We thought it worthwhile to clarify this, as the PD12 model can include an isostatic adjustment option, which has been used in previous studies.

The bed is constant in time, and has an elevation of $-100$ m at the divide and constant prograde slope $b_x$ of $-2 \times 10^{-3}$

*167 "S\cross L" should be "S\cdot L" or just "SL". Do not put variable names in paren- theses in some places.*

$S \times L$ has been replaced with $S \cdot L$ throughout.

*167 "Q" : what is this, the flux into the grounding zone? Please make description clearer.*

Yes – this has been clarified:

Ice thus enters the system via an accumulation flux $S \cdot L$, flows via an interior flux $Q$ to the grounding zone, and leaves the system as a flux across the grounding line $Q_g$ .

*190 Repeat which variable is the "mass balance rate" (I think S)*

Yes it is $S$ – this has been clarified.

*192 What is this "small reservoir"? Until now there was just one, described by S and L. Maybe this is the purple box in Figure 2. If so, this setup should be made more explicit from the beginning, and Figure 2 should be improved.*

Yes, it is the grounding zone reservoir and the purple box in figure 2. We have clarified here that it refers to the grounding zone:

$\tau_F$ is controlled by hg, and thus the volume of the system's small grounding-zone reservoir

However, the two reservoirs are already described above this, and the figure referenced: "The glacier can be conceptualized as a small grounding-zone reservoir with a length lgz ≪ L and thickness hg , coupled to a large interior reservoir with thickness H and length L−lgz. … The model dynamics are derived by balancing ice fluxes through these linked reservoirs (Fig. 2a)."

*200 It is not clear what the use of a linearized system is. It is certainly useful to find eigenvalues and eigenvectors at a certain state.*

*But having a "full" model with all nonlinearities, and then using a linearized "deviation model" makes no sense for the large changes investigated. Also, the "full" model is extremely cheap to calculate, as it is just a 2-variable ODE.*

*Especially given the comments in line 210 that the nonlinear ("full") ODE give the same results makes one wonder, why the linearized version is used at all (except for investi- gation of eigenvalues).*

This is an important question, and we are glad it has been raised. We view the linear model primarily as a tool for analysis. As noted in this comment, it has eigenvalues (here, $1/\tau_s$ and $1/\tau_S$), which are key to understanding the transient response. In our view, the eigenvalues are just one benefit of the linear model. Linear frameworks are fundamental in timeseries analysis and statistics, and allow one to apply a broad and well-established toolkit to analyze the system in question. For example, approximating the outlet glacier as a linear system not only allowed Robel et al. (2018) to identify that there are two characteristic timescales, but also provided a framework that shows how they operate relative to each other (i.e., their relative contributions to the total system response). In the present study, we find this framework to be helpful for understanding how the same two response times can yield different behavior depending on how forcing is applied.

Perhaps most importantly, the linear response times reveal how physical parameters govern the transient response. Even if a particular linearization is only strictly valid for small changes, these leading controls have robust physical interpretations, and so are useful for understanding the system. These insights would not be as readily available for response times estimated empirically from more complex models or data.

Of course, there are many salient aspects of marine-terminating glacier dynamics that are fundamentally nonlinear, which can't be fully addressed with linearized models. However, as the *physical* controls identified by linearization often translate to some degree into the nonlinear cases, linear models can help point to questions to be tested in a more complex model. Our point is not that the linear model is superior, but it does enable understanding in a unique way.

An additional and more specific point raised in this comment is why the linear model should be used (especially for large changes) when the nonlinear version is available and computationally cheap. The limit of a linearization is an important consideration, which we discuss further below

in response to a related comment. However, in this section, we think it is in fact useful to compare the linearized response to the flowline model, as it reveals just how nonlinear the response is. For the relatively large forcing imposed (20% perturbations in S and Ω), there are indeed some differences between linear and flowline sensitivities (and as noted in the text, the nonlinear two-stage and flowline models have identical *equilibrium* sensitivity). However, the key transient aspects stand out in light of these differences. We feel it is important to show the linear response here in order to back up the later experiments and interpretations that are based on the linear framework. However, we agree that this motivation was not clear enough in the original text. We have re-worked the following paragraph in order to clarify why the linear output is shown here, and also to motivate its use later in the paper:

Figure 2 shows output from both models in response to step and stochastic forcings. Fig. 2b shows the grounding line retreat following a 20% increase in Ω (blues) and 20% decrease in S (orange/brown). For clarity, only the linearized two-stage output is shown. Note that the nonlinear two-stage model is constrained to have the same equilibrium response as the flowline model by Eq. (1). The linear and nonlinear responses match almost exactly for the stochastic fluctuations, but disagree somewhat for the step changes. However, the disagreement is not severe, suggesting that we can reasonably use the linearized model and its response times as analytical tools. Importantly, the two-stage model captures the faster initial response to forcing at the grounding line, with a slightly more pronounced slope break in the first few hundred years.

*250 Sn important additional argument in this discussion is also the spatial scale. Interior changes take a certain time, given by the ice flow speed, to affect the terminus, while processes at the terminus are immediately affecting glacier length. This is the same as on any glacier, but the effect of terminus dynamics in tidewater are much faster and bigger than those of terminus melt on a mountain glacier.*

We agree that this is an important consideration. We believe the essence of this basic point is already brought up in the next paragraph:

"A change in surface mass balance has an immediate, spatially distributed tendency on interior thickness, which slowly alters driving stresses and thus fluxes throughout the interior. Anomalous ice flux arrives at the grounding zone … In contrast, a perturbation to $Q_g$ (here, via Ω) is highly localized and first creates a large disequilibrium between $Q_g$ and interior flux…"

A related issue is the spatial pattern of the mass balance forcing. In the discussion, we bring up this issue and describe the effects of localizing the surface perturbation near the terminus.

*265 Here a link to kinematic wave speed would be very interesting. How fast is the terminus signal propagating upstream.*

This is an interesting point and a useful connection to make. The signal propagating up-glacier is diffusive, so we must define some threshold to track propagation. A natural one in this case is the peak in interior fluxes, shown in Figure 3. This shows when the gradual drawdown of interior ice (a negative tendency on flux) overwhelms the initial increase allowed by the perturbation at the grounding line. We now discuss this in this paragraph on flux evolution. We would prefer to discuss its speed in terms of the fast timescale (rather than absolute numbers), since this

paragraph is intended to be a general and qualitative comparison of the flux changes between the two models.

"… Both models capture this transfer, and the flowline model flux gates show that it gradually propagates up from the grounding zone. The transient peak in fluxes is similar to a kinematic wave propagating from the terminus, which reaches well into the interior within a few multiples of $\tau_F$ . The ensuing drawdown of the interior reservoir (the slow mode in the two-stage model) brings interior fluxes back down, and again, $Q_g$ must follow via grounding line retreat.

*Figure 3: The two blueish-greenish colors are difficult to distinguish, use a better color table.*

Fixed. We have used higher-contrast colors and bolder lines. Example:

[Figure]

*Figure 4: Use clearer colors, full red, blue etc. They might look less appealing, but are easier discernible.*

We have made the reds and blues brighter. We would prefer to retrain the scheme where the lighter hues (and bolder lines) of the same color (black, red, maroon, blue) correspond to forcing with the same spectral characteristics, but applied as S anomalies. We tried other options to distinguish (like dashed lines), but found these to be less visually clear.

Example:

[Figure]

*308 "white noise forcing" (also 311)*

Fixed.

*346 where do these exact numbers come from?*

These are the linearized response times predicted by the two-stage model. Table 2, which is referenced shortly after, contains the other parameters for each glacier. We have added to the table 2 caption to clarify how these values are determined:

Table 2. Parameters varied between three idealized glaciers (top) and the resulting steady-state values (bottom). The steady state is calculated by the nonlinear two-stage model, and the linearized response times are given by Eqs. 13 and 14.

*350 should the "as of 2020" follow "transient response"?*

Yes – thank you for catching this. Clarified:

The most basic result is that, in all cases, the transient response as of 2020 is a small fraction (~ 1–23%) of the instantaneous equilibrium response.

*Figure 5: caption: "Table" (upper case)*

Fixed.

*457 Here the question posed above gets more pressing: why is the linearized model used? Can it be used at all for such large changes? How wrong do the results get? Why is not the "full" simple model used here?*

(see full response after next comment)

*458 And then also: are the time scales always the same, even if the glacier geometry changes by a very large amount?*

We will address the above two related comments together. These are good points and valid concerns for applying the linear model to the question of a long-term memory of past climates. We would agree that a linear model probably cannot accurately capture such large changes (i.e., Last Glacial Maximum to Holocene), and would not be the right choice for reconstructions.

The main reason we use the linear model here is that it is constrained to have the same equilibrium sensitivity to fractional anomalies in S and Omega. This allows us to more easily compare the transient responses side-by-side. Differences in the transient responses at any given time are thus due only to the different expression of fast and slow responses following each type of forcing.

One could use the nonlinear two-stage model (or the flowline model) here and normalize the results, potentially capturing some relevant nonlinearities. However, some of the nonlinearities depend on simplifications still present in the nonlinear models. Furthermore, their effects would depend on the magnitude of the forcing applied. For this highly idealized case (and one set of model parameters), we would be concerned to add model complexity that might overemphasize effects that vary widely between real settings. (For example, one nonlinearity inherent in these models is that glacier advance expands the surface area, increasing total accumulation flux. This would vary between real settings, and would be reversed in the presence of an ablation zone.)

Further study on various nonlinearities and how they vary between settings would of course be an interesting and very useful topic for future research.

We do not mean to argue here that the linear equations are a superior model. For this experiment, however, we believe it is more straightforward to (a) demonstrate a few simple points; and (b) to explain its assumptions and limitations.

We agree that the original manuscript should have been clearer, particularly on (b). We thus have tried to highlight the reasons for using the linear model, and, importantly, to shift emphasis to the smaller climate variations (e.g., LIA and late-Holocene cooling) for which the linear model is more valid. We think these variations are quite interesting in their own right, so we are glad that this comment gave us an opportunity to sharpen the focus of this section.

The revised part of this section is below, with new or modified statements in blue.
* * *
First, we consider a climate scenario with idealized representations of three events: a deglacial warming at 11 kya, a Little Ice Age (LIA) cool period from 1450 to 1850 CE, and an anthropogenic warming trend from 1880 to 2100 CE. The deglacial and LIA transitions are smoothed by error functions of 500 yr and 50 yr widths, respectively. The magnitudes of deglacial, LIA, and anthropogenic events have a ratio of 10:1:4, and the intervening Holocene climate is assumed constant. We scale forcings linearly to the climate anomalies, where warming corresponds to negative $S'$ or positive $\Omega'$. Obviously, different combinations of these forcings could be expected in reality. Rather than choosing a particular combination, we examine each in isolation and normalize the glacier responses. These experiments thus serve as limiting cases to illustrate the relative influences of ocean and interior forcing over different timescales. This scenario is designed to explore two practical points: (1) the glacier's memory of large, long-ago events compared to smaller, more recent events; and (2) the glacier's relative memory of past ocean vs. interior forcing.

We use the two-stage model, linearized with respect to the Holocene climate with parameters for glacier 1 (See Table 2; $\tau_F$, $\tau_S \sim 76$, 2000 yrs). The advantage of using the linear model here is that it has uniform sensitivity to fractional perturbations in $S$ and $\Omega$. This allows us to more clearly distinguish the signatures of fast and slow dynamics; the tradeoff is that it ignores nonlinearities that are surely a factor for very large climate changes. Accurate simulations over such transitions would depend not only on nonlinearities in ice dynamics, but also on spatial information (e.g., bed topography) that is simplified in reduced models. Nevertheless, the linear model is a straightforward tool for demonstrating consequences of having both fast and slow dynamics. Focus should be directed toward the fast and slow responses to the idealized LIA, for which the linearization is more valid. Including the deglaciation signal primarily serves to account for residual, albeit faint, disequilibrium implied by millennial response times.

Figure 7a shows the idealized climate (top) and two-stage model responses. Anomalies are shown relative to the mid- Holocene and normalized to the large deglacial transition. Figure 7b shows 1000 to 2000 CE in more detail, including a scenario with no anthropogenic warming (dashed). For reference, the length at which the glacier would be in equilibrium with the LIA climate is also plotted (gray). For both forcings, the grounding-line retreat due to the deglacial signal is nearly complete by the onset of the LIA. However, the LIA advance is $\sim 2\times$ greater for forcing in $\Omega$, because it can engage the fast mode to a greater degree. Yet, even forcing in $\Omega$ yields a muted response: the transient response only reaches $\sim 35\%$ equilibration before the period ends. The slow mode, which takes up the majority of the response for both forcing types, barely feels our 400 yr LIA before it is reversed. This is worth bearing in mind whenever the duration of glacier excursions and climate anomalies are less than $\tau_S$, because the system never achieves equilibrium. This would be an issue particularly if such events are used to tune glacier sensitivity in models.

The idealized LIA is a much more discrete "event" than is supported by paleoclimate records, and ignores other variations in the Holocene. Thus, we also consider the response to a more realistic forcing time series. Again, this is not a reconstruction of actual terminus changes, but it is useful to see how glacier

memory integrates the continuum of variations found in paleoclimate records. We use a time series of temperatures for Disko Bay, Greenland, from the regional reconstruction of Buizert et al. (2018)
* * *
*467 "time series" (two words)*

Corrected throughout.

**References in responses**

Goldberg, D. N., Heimbach, P., Joughin, I., and Smith, B.: Committed retreat of Smith, Pope, and Kohler Glaciers over the next 30 years inferred by transient model calibration, The Cryosphere, 9, 2429–2446, https://doi.org/10.5194/tc-9-2429-2015, 2015.

Haseloff, M. and Sergienko, O. V.: The effect of buttressing on grounding line dynamics, J. Glaciol., 64, 417–431, https://doi.org/10.1017/jog.2018.30, 2018.

Joughin, I., Smith, B. E., Howat, I. M., Floricioiu, D., Alley, R. B., Truffer, M., and Fahnestock, M.: Seasonal to decadal scale varia- tions in the surface velocity of Jakobshavn Isbrae, Greenland: Observation and model-based analysis, J. Geophys. Res. Earth Surf., 117, https://doi.org/10.1029/2011JF002110, 2012b.

Nick, F. M., Vieli, A., Howat, I. M., and Joughin, I.: Large-scale changes in Greenland outlet glacier dynamics triggered at the terminus, Nat. Geosci., 2, 110–114, https://doi.org/10.1038/ngeo394, 2009.

Price, S. F., Payne, A. J., Howat, I. M., and Smith, B. E.: Committed sea-level rise for the next century from Greenland ice sheet dynamics during the past decade, Proc. Nat. Acad. Sci. USA, 108, 8978–8983, https://doi.org/10.1073/pnas.1017313108, 2011.

Robel, A. A., Roe, G. H., and Haseloff, M.: Response of Marine-Terminating Glaciers to Forcing: Time Scales, Sensitivities, Instabilities, and Stochastic Dynamics, J. Geophys. Res. Earth Surf., 123, 2205–2227, https://doi.org/10.1029/2018JF004709, 2018

Roe, G. H. and Baker, M. B.: Glacier response to climate perturbations: An accurate linear geometric model, J. Glaciol., 60, 670–684, https://doi.org/10.3189/2014JoG14J016, 2014.

Schoof, C.: Ice sheet grounding line dynamics: Steady states, stability, and hysteresis, J. Geophys. Res. Earth Surf., 112, https://doi.org/10.1029/2006JF000664, 2007a.

***KEY***

*Reviewer comments (green italic)*

Response (black)

New or changed text (blue)

**Anonymous Referee #2**

*In this study, the authors used simplified models to explore the response of ice flow to different sources of climate forcing, including perturbations at the grounding line and to the surface mass balance in the ice interior. I found this approach to be novel and very interesting. It successfully provides new insights into the connections between various forcings, in terms of amplitude and effects operating at different timescales.*

*The manuscript is well organised and very well written, overall. I support its publication after minor corrections.*

Thank you for the encouraging review, and for suggestions that we think have led to a clearer manuscript. Please see our responses and changes below.

*Comments:*

*-L. 93: delete "model" in "The PD12 model model. . ."*

Fixed.

*-L. 203: It isn't clear to me, what does "a different flowline model" refer to. Is it simply the flowline model described in section 2.2?*

Thank you for catching this ambiguity. Robel et al. used a different flowline model to the one described here, which we now note more explicitly:

RRH showed that the two-stage model emulated the response of a flowline model forced with surface mass balance anomalies. Their flowline model (described in Schoof, 2007b; Robel et al., 2014) used a stress-based, as opposed to flux-based, grounding-line condition, but was otherwise dynamically similar to the PD12 model.

*-L. 339: correct "instantaneous equilibrium"*

Fixed.

-L. 372, and first paragraph after subtitle: it would be helpful to define "emergence and detectability" more explicitly. As currently presented, I am not sure how the paragraph (lines 373-377) introduces the section.

This paragraph is meant to establish up front that we are focusing specifically on the challenges posed by transient glacier dynamics. We agree that this may have been too abstract as written to properly introduce the section, and have made the following changes to clarify this:

3.2 The emergence and detectability of forced responses

We now turn to the topic of attributing outlet glacier retreats to natural or anthropogenic forcing. The attribution of an observed change to a particular cause (i.e., an external forcing) can be a challenge because of factors specific to individual glaciers, such as complexities in bed geometry, regional climate, or the local collection of ice-ocean interactions. It can also be a challenge because of factors intrinsic to the transient ice dynamics, which affect the amount of the forced response that can be expressed over a given time. We focus here on this latter set, and in particular on the contrasting implications of ocean vs. interior forcing.

-L. 381 and 382: Can you clarify how the glaciers' memory mentioned relates to the committed change discussed previously?

Great point to tie this to the previous section. We now reference Figure 5 on committed change.

-L. 386: Specify which "two types of forcing" you are talking about (ocean vs interior, presumably?)

Yes – clarified:

…two types of forcing (ocean and interior) and much longer response times…

-L. 399: can you specify "detectability. . ." of what?

Yes – we have re-worded to clarify:

…the very slow response to the trend in S means that the forced response is slow to emerge from the noise; it remains within $2\sigma_L$ until the late 21st century.

-L. 458: Did you mean a reference to Figure 7c?

Yes – fixed.

Generally, I find the figures to be clear, although I would suggest working a bit more on / completing some of the figure legends, in particular for Figure 1, Figure 2, Figure 5 and Figure 6. For example:

Figure 1: insert "(blue)" and "(orange)" after "omega" and "S" respectively, in caption d.

Fixed.

Fixed.

Figure 5: Caption needs to be more precise: aren't panels a, b and c showing the response of glaciers to idealized climate forcing? As it stands, it reads as if they show the climate forcing itself.

Agreed this was unclear. We have re-organized to clarify:

Glacier responses to idealized climate forcing over the industrial era. (a) Responses for glacier 1…

Figure 6: Similarly, some details and descriptions are missing. E.g., suggest completing the legend for panel b. I would also suggest making the titles currently in grey stand out a bit more.

Fixed. The figure and caption have been revised to be more descriptive:

[Figure]

Figure 6. Detecting the response to a climate trend in the presence of natural variability. Three types of natural variability are considered in each column. The top row corresponds to interannual variability in S; middle row to interannual variability in $\Omega$; and bottom row to multidecadal variability ($\tau_{AR1} = 20$ yr) in $\Omega$. (a) The idealized climate trends (plus variability) 
[revised manuscript text omitted]

 We assume a typical power-law rheology, with coefficient $A$ and exponent $n$  (e.g., Glen, 1955). Internal shear deformation ($ \frac{\partial u}{\partial z}$) is represented by the shallow-ice approximation. At a depth $h-z$, this is

$$\frac{\partial u}{\partial z}=2A\tau_e^{n-1}\rho_i g(h-z)\frac{\partial s}{\partial x} \tag{6}$$

where $ \tau_e^2=(\frac{1}{2})\tau_{ij}\tau_{ij}$ is a scalar effective stress and $\tau_{ij}$ is the deviatoric stress tensor. Finally, sliding velocity is given by a Weertman-type power-law relationship

$$u_b= \left(\frac{\tau_b}{C}\right)^{\frac{1}{m}}, \tag{7}$$

where $C$ is a friction coefficient and $m$ is the sliding exponent. Here, $m$ is the inverse of the flow exponent $n$, but note that this convention is flipped in some texts. The PD12 model  uses a combination of the shallow-ice and shallow-shelf approximations to solve for $ \frac{\partial u}{\partial z}$  and $\frac{\partial u}{\partial x}$, respectively. In this study, however, we use parameters that yield flow dominated by longitudinal stretching. We refer the reader to PD12 for further description of this hybridization.

A crucial simplification in the PD12 model is that flux across the grounding line is parameterized in the form of  Equation (1). In PD12 and this study, $\Omega$ and $\beta$ are taken from the analytical solution of Schoof ( 2007a):

$$\Omega=\left[A(\rho_i g)^{n+1}\left(\Theta\left(1-\rho_w/\rho_i)\right)\frac{\rho_w}{\rho_i}\right)\right)^n(4^nC)^{-1}\right]^{\frac{1}{m+1}} \tag{8}$$

$$\beta=\frac{m+n+3}{m+1}. \tag{9}$$

$\Theta$ is a buttressing factor between  0 and 1, and all other parameters are defined as above. The PD12 model calculates grounding line flux based on a thickness  that is linearly interpolated from the height above flotation of the last-grounded and first-floating grid cells, to the point where flotation is reached. The corresponding sub-grid grounding-line position is shown for all output from the PD12 model in this study. Although the PD12 model is typically run on coarse grids ($ \mathcal{O}\sim1\text{–}10$ km) for continent-scale simulations over many millennia, we use a grid of  100 m to better resolve the details of grounding line variations.  Finally, while the PD12 model does simulate a floating ice shelf, its dynamics are not important to our analyses as its buttressing effect is parameterized via  Equation (8).

Figure 1b shows the steady-state profile for an idealized outlet glacier simulated with the flowline model. The domain begins at an ice divide and thus has a zero-flux lateral boundary condition. The bed is constant in time, and has an elevation of $-100$

m at the divide and constant prograde slope ($b_x$ of $-2 \times 10^{-3}$. Surface mass balance ($S$ is 0.5 m yr$^{-1}$ ice equivalent, assumed to be spatially uniform. Additional parameters are given in Table 1. The glacier has an equilibrium length (ice divide to grounding line) of $\sim 185$ km, a maximum thickness of $\sim 1580$ m at the divide, and a thickness of $\sim 526$ m at the grounding line. These scales are comparable to many outlet glacier catchments in Greenland, though direct comparisons are limited as the 1-D flowline cannot capture the flow convergence of many catchment geometries, and we emphasize that we are not simulating a particular outlet glacier. We discuss two additional geometries for comparison in Section 3.1.

**2.3 Response to forcing**

We begin by comparing the flowline model's response to forcing either from surface-mass-balance changes in the interior or ocean forcing at the terminus. In all model experiments throughout this study, "interior forcing" and "ocean forcing" will refer to perturbations applied in the following manner. Interior surface-mass-balance anomalies are assumed to be spatially uniform. We represent ocean forcing very simply by perturbing the grounding-line-flux coefficient, $\Omega$. This broadly represents changes in the buttressing provided by an ice shelf or fjord walls, driven by anomalous melting or calving. In principle this could be targeted via $\Theta$ in Equation (8), and other analytical formulations for $\Omega$ explicitly represent a buttressing ice shelf (Haseloff and Sergienko, 2018). Perturbing these parameters might be more realistic, but we focus on $\Omega$ so that we can very generally represent flux perturbations, which may in reality result from a host of ice-ocean interactions. Our primary interest is in how glacier dynamics respond to each forcing type, and representing ocean forcing in this way makes comparison straightforward. For example, a fractional change in $S$ (surface mass balance) constitutes a flux anomaly with the same initial magnitude as the same fractional change in $\Omega$.

Figure 1c shows the length response to step forcings at time $t = 0$. The step is a 20% decrease in $S$ or a 20% increase in $\Omega$ (i.e., more discharge for a given $h_g$), and in both cases, forces the terminus to retreat into shallower water to reduce the output flux. Note that the responses to changes in $S$ and $\Omega$ do not converge to the same final length. The equilibrium sensitivities differ because of different nonlinearities in the input and output fluxes ($S \cdot L$ and $Q_g$) as the system state evolves. However, our main focus is the marked difference between the initial responses to the step change, highlighted in the inset panel of Fig. 1c. Perturbing $\Omega$ results in a much faster initial retreat (blue curve) compared to perturbing $S$ (orange curve), despite the larger equilibrium sensitivity to $S$. However, this faster retreat rate lasts only the first century or so, and accommodates only $\sim 25\%$ of the total equilibrium response. The remaining retreat occurs at a rate similar to that driven by a change in $S$, which is an asymptotic approach to equilibrium over several millennia. Thus, the length response to $\Omega$ forcing has both a "fast" and "slow" component, whereas $S$ forcing primarily drives a slow response.

As an alternative to an impulse forcing, we also consider the glacier's response to stochastic variability in either $\Omega$ or $S$. We apply this variability as interannual white noise, which by definition has equal power at all frequencies and no interannual persistence. We apply the exact same white-noise time series as either $\Omega$ or $S$ anomalies, but with opposite sign so that the corresponding ice-volume anomalies match. The anomaly time series (hereafter $\Omega'$ and $S'$) are scaled to have standard deviations equal to 20% of the mean values ($\bar{\Omega}$ and $\bar{S}$). Figure 1d shows the resulting length responses. For both types of forcing, the glacier acts as a low-pass filter on the imposed climate anomalies,

**Table 1.** Parameters used for flowline and two-stage models. Values for $A$ and $C$ follow previous idealized case studies (Schoof, 2007a; Robel et al., 2018).

| Parameter | Description | Value | Units |
|---|---|---|---|
| $m$ | Sliding exponent | $\frac{1}{3}$ | |
| $C$ | Sliding coefficient | $7.624 \times 10^6$ | Pa m$^{-\frac{1}{3}}$ s$^{\frac{1}{3}}$ |
| $n$ | Deformation exponent | $3$ | |
| $A$ | Deformation coefficient | $4.22 \times 10^{-25}$ | Pa$^{-3}$ s$^{-1}$ |
| $\Theta$ | Buttressing factor | $0.7$ | |
| $\rho_w$ | Seawater density | $1028$ | kg m$^{-3}$ |
| $\rho_i$ | Ice density | $917$ | kg m$^{-3}$ |
| $\bar{S}$ | Surface mass balance | $0.5$ | m yr$^{-1}$ |
| $b_0$ | Bed elev. at divide | $-100$ | m |
| $b_x$ | Bed slope | $-2 \times 10^{-3}$ | |

producing kilometer-scale fluctuations with clear persistence. However, the length anomalies driven by $\Omega'$ have much greater high-frequency content, and greater overall variance, than those driven by $S'$. The high-frequency response is perhaps intuitive, in that forcing applied at the grounding line has an immediate effect on grounding-line position. However, the response to $\Omega'$ also contains millennial-scale fluctuations onto which the faster variations are superimposed. These slow, wandering excursions are comparable to those driven by $S'$, and, like the multi-millennial adjustment to step changes (Fig. 1c), suggest slow dynamics common to both responses.

Variability is intrinsic to climate, arising from the fundamentally chaotic nature of the natural system. The associated glacier fluctuations are a crucial part of characterizing glacier dynamics, and the implications have been extensively explored for mountain glaciers (e.g., Oerlemans, 2001; Roe, 2011; Roe et al., 2017), and more recently for marine-terminating outlet glaciers (Robel et al., 2018) and ice streams (Mantelli et al., 2016). Figure 1 suggests a new and intriguing complication for outlet glaciers: the timescale and magnitude of glacier fluctuations depend on whether the climate variability comes in the form of surface mass balance or ocean anomalies. Additionally, the response to step changes (Fig. 1c) suggests that the type of forcing is also relevant for the response to non-stationary climate changes. In the next section, we investigate these contrasting responses using a recently developed reduced model.

**2.4 The two-stage model**

Robel, Roe, and Haseloff (2018; hereafter RRH) developed a simple model for marine-terminating outlet glacier dynamics, derived from ice-flux conservation and constrained by large-scale glacier geometry. RRH showed that this model could accurately emulate a more complex numerical model on the multidecadal and longer timescales on which most glacier variance occurs. A full description and derivation can be found in RRH, but a summary of the model follows.

[Figure]

**Figure 1.** Model schematic and  responses to ocean forcing (via $\Omega$) and interior forcing (via $S$). **(a)** An idealized marine-terminating glacier system. **(b)**  Initial steady-state profile  simulated by the flowline model. **(c)** Length  responses to  20% step increase in $\Omega$ (blue) and  20% step decrease in $S$ (orange) at $t = 0$. Inset panel zooms into the first  0.5 kyr of response. Note the difference in initial retreat rates depending on the type of forcing. **(d)** Length  responses to  white-noise interannual variability in $\Omega$  or $S$. Right panel shows a 0.5 kyr segment in more detail. Anomalies (black) have a standard deviation of 20% of the mean value, and have opposite signs for $\Omega$ and $S$ so that length anomalies  are correlated for comparison. The response to ocean forcing  ($\Omega$blue) contains much more high-frequency content than the response to interior forcing (orange).

The RRH model reduces the outlet glacier to a system with two degrees of freedom: $L$, the length from ice divide to grounding line; and $H$, the  characteristic interior ice thickness (Fig. 2a). The glacier can be conceptualized as a small grounding-zone reservoir with a length $l_{gz} \ll L$ and thickness $h_g$, coupled to a large interior reservoir with thickness $H$ and length $L - l_{gz}$.  The geometry is further described by a  bed topography with constant average slope $b_x$. Recall that the ice thickness at the grounding line, $h_g$, is a state-dependent quantity, set entirely by $L$ and $b(x)$ (see Eq. 2).

The  model dynamics are derived by balancing ice fluxes through these linked reservoirs (Fig. 2a). Ice thus enters the system via an accumulation flux  $S \cdot L$, flows via an interior flux $Q$ to the grounding zone, and leaves the system as a flux across the grounding line . A steady state of $H$ and $L$ is one which balances these three fluxes. $L$ ultimately controls the system's input and output fluxes by setting the total catchment area for accumulation, and by controlling $Q_g$ via $h_g$ (Eqs. 1 and 2). Interior dynamic ice flux has the form

$$Q = \left(\frac{\rho_i g}{C}\right)^n \frac{H^\alpha}{L^\gamma}, \qquad (10)$$

where $\alpha$ and $\gamma$ can vary depending on the particular processes controlling interior flow. We use $\alpha = 2n + 1 = 7$ and $\gamma = n = 3$, consistent with deformation dominated by longitudinal stretching (RRH).

Two coupled equations capture the transient adjustment of the two degrees of freedom, $H$ and $L$, as they relax towards a steady state that balances all three fluxes:

$$\frac{dH}{dt} = S - \frac{Q}{L} - \frac{H}{h_g L}(Q - Q_g) \tag{11}$$

$$\frac{dL}{dt} = \frac{1}{h_g}(Q - Q_g). \tag{12}$$

Because achieving steady state requires adjustment of both $H$ and $L$, we refer to this model as the "two-stage model", following RRH. We discuss the operation of these stages further in Section 2.6. Note that Equation (10) captures the nonlinear dependence of ice flux on driving stress (e.g., Eqs. 4–6) and ice thickness, and that the first two terms of  Equation (11) resemble the continuity equation that governs ice thickness in the flowline model (Eq. 3). The two-stage model makes significant simplifications, but stems from the same physical principles as the flowline model, and most other contemporary ice-sheet models (see RRH for further discussion).

RRH also linearized these equations for fluctuations $H'$ and $L'$ about a steady state, $\bar{H}$ and $\bar{L}$. The linear model yields analytical expressions for a number of important quantities, including two widely separated characteristic response timescales. With some simplifications detailed in RRH, the fast and slow response timescales ($\tau_F$ and $\tau_S$, respectively) are

$$\tau_F = \frac{\bar{h}_g}{\bar{S}}(\alpha + \gamma + 1 - s_T)^{-1} \tag{13}$$

$$\tau_S = \frac{\bar{H}}{\bar{S}s_T\alpha}(\alpha + \gamma + 1 - s_T), \tag{14}$$

where $s_T = 1 + \frac{\rho_w}{\rho_i}\frac{\beta\bar{b}_x\bar{L}}{h_g}$ is a parameter describing the stability of glacier response based on its geometry and grounding-line dynamics (via $\beta$). Both response times contain a characteristic thickness divided by mass balance rate $\bar{S}$, reminiscent of the canonical mountain glacier response time (Jóhannesson et al., 1989). $\tau_F$ is controlled by $h_g$, and thus the volume of the system's small grounding-zone reservoir, whereas $\tau_S$ relates to the volume of the interior reservoir via $\bar{H}$. For scales typical of outlet glaciers, $\tau_F$ is on the order of decades to centuries, whereas $\tau_S$ is on the order of millennia. Thus, the two-stage model approximates the outlet-glacier system as linked interior- and grounding-zone reservoirs with distinct timescales.

Here we generalize the RRH linearization to simultaneously include perturbations to interior surface mass balance ($S'$) and grounding-line flux ($Q_g'$; proportional to $\Omega'$). The linearized equations take the form of a two-dimensional dynamical system with two forcing terms:

$$\frac{\partial}{\partial t}\begin{bmatrix} H' \\ L' \end{bmatrix} = \begin{bmatrix} A_H & A_L \\ B_H & B_L \end{bmatrix}\begin{bmatrix} H' \\ L' \end{bmatrix} + \begin{bmatrix} \bar{L}^{-1}\left(\frac{\bar{H}}{h_g} - 1\right) \\ \bar{h}_g^{-1} \end{bmatrix}Q_g' + \begin{bmatrix} 1 \\ 0 \end{bmatrix}S'. \tag{15}$$

$A_H$, $A_L$, $B_H$, and $B_L$ are shorthand for the couplings between length, thickness, and flux changes, which are given in the appendix. This linear system is readily discretized, and can be cast as a two-dimensional autoregressive process  (e.g., Box et al., 2008), which we also present in the appendix.

**2.5 Model comparison**

RRH showed that the two-stage model emulated the response of a  flowline model forced with surface mass balance anomalies. Their flowline model (described in Schoof, 2007b; Robel et al., 201 used a stress-based, as opposed to flux-based, grounding-line condition, but was otherwise dynamically similar to the PD12 model. Here, we use the PD12 model and extend the comparison to grounding-line flux perturbations. Although the two-stage model output includes $H$, we focus our comparison on $L$ anomalies, which are more directly comparable between models. $H$ is a more heuristic variable in the two-stage model; however, it does govern the evolution of interior fluxes, which we discuss in the next section. Both models use the parameters given in Table 1, for which Equations (13) and (14) give response times of $\sim 76$ and $\sim 2000$ years, respectively.

Figure 2 shows output from both models in response to step and stochastic forcings. Fig. 2b shows the grounding line retreat following a 20% increase in $\Omega$ (blues) and 20% decrease in $S$ (orange/brown). For clarity, only the linearized two-stage output is shown.  Note that the nonlinear two-stage model is constrained to have the same equilibrium response as the flowline model by  Equation (1).  The linear and nonlinear responses match almost exactly for the stochastic fluctuations, but disagree somewhat for the step changes. However, the disagreement is not severe, suggesting that we can reasonably use the linearized model and its response times as analytical tools. Importantly, the two-stage model captures the faster initial response to forcing at the grounding line, with a slightly more pronounced slope break in the first few hundred years.

Figure 2c shows a  5 kyr sample of  100 kyr of length fluctuations driven by white-noise forcing in $S'$ and $\Omega'$. Again, anomalies have no interannual persistence, are identical except for a sign reversal between $\Omega'$ and $S'$, and have a standard deviation of  20% of $\bar{\Omega}$ or $\bar{S}$. The two-stage and flowline models agree closely for both types of forcing, although the two-stage model slightly underestimates the magnitude of high-frequency fluctuations. Figure 2d shows the autocorrelation function of each length  time series. This is a measure of the persistence of anomalies, and reveals the memory within a dynamical system. For forcing from $\Omega'$, the autocorrelation drops off rapidly to approximately $0.4$ after a couple of centuries, corresponding to the fast response time. However, it then requires several millennia to drop below $\sim 0.1$, indicating the persistent influence of the slow timescale. For $S'$ forcing, the autocorrelation is dominated by the slow timescale. The power spectra of length fluctuations are shown in Fig. 2e. Because white-noise forcing has a flat power spectrum, the shapes of the response spectra reveal the glacier's filtering properties. The glacier is a strong low-pass filter for both forcing types, but damps high frequencies even more for $S'$ forcing. The two-stage model underestimates the high-frequency response of the flowline model, but agreement is very good at the lower frequencies that contain the majority of the variance. Critically, both

[Figure]

**Figure 2.** Comparing  transient  response of  flowline and two-stage models to ocean forcing (via $\Omega$) and interior forcing (via $S$). **(a)** Schematic of the two-stage model. **(b)**  Flowline and linearized two-stage model length responses to step changes in $\Omega$ (blues) and $S$ (orange/brown). The thinner lines show the two-stage output. **(c)** Length anomalies in response to white-noise forcing  in $\Omega$ (blues) and $S$ (orange/brown). As in  Figure 1, anomalies have standard deviation of  20% of the mean value and have opposite signs.  5 kyr of the  100 kyr model runs are shown. **(d)** Autocorrelation function for the stochastic length fluctuations shown in **(c)**. **(e)** Power spectral density for the stochastic length fluctuations, which is estimated throughout this study via Welch's method with windows of  1/16 the  time series length, and  50% overlap. Note the clear split between fluctuations driven by $\Omega'$ (blues) and $S'$ (orange/brown) at millennial and shorter timescales. This split is robust across models. The flowline model has more high-frequency power for both types of forcing, but note that the spectral power is orders-of-magnitude lower at such high frequencies.

models capture the clear split between the spectra of $\Omega'$- and $S'$-driven anomalies at frequencies around $10^{-2}$ to $10^{-3}$ yr$^{-1}$, where the models agree quite well.

**2.6 Interpretations**

The two-stage model's agreement with the flowline model suggests that it captures the essential dynamics responsible for the contrasting transient responses to interior and grounding-line flux anomalies. At this point, it is useful to discuss some of the interpretations enabled by this reduced model.

The linearized two-stage model (Eq. 15) is a dynamical system with two eigenmodes, which, for the stable geometries considered here, are exponentially decaying functions with characteristic times $\tau_F$ and $\tau_S$ (the eigenvalues being $\tau_F^{-1}$ and

$\tau_S^{-1}$). The responses of the system's two degrees of freedom ($H'$ and $L'$) are linear combinations of these "fast" and "slow"
eigenmodes. However, the two forcing types, $\Omega'$ and $S'$, project onto each mode with different relative weights. Forcing via $S'$
projects almost entirely onto the slow mode. For forcing in $\Omega'$, the fast mode makes a substantial contribution on multi-decadal
to centennial timescales. However, the slow mode still dominates the equilibrium response to an impulse, as well as the power
spectrum of stochastic fluctuations.

These modes can also be conceptualized as a  two-stage low-pass  filter on any forcing time series (e.g., Fig.
2e), and the type of forcing determines the order in which they are applied. Forcing over the interior ($S'$) is first filtered by the
slow mode, and the fast mode then operates on interior flux anomalies whose high-frequency content has already been strongly
damped; there is little additional filtering to do. In contrast, if forcing is applied at the grounding line ($Q_g'$ via $\Omega'$), the fast mode
responds to unfiltered anomalies and the grounding-line position exhibits a fast initial response before engaging the slow mode
in the interior. Figures 1 and 2 suggest that the response to $S'$ could be reasonably approximated by the adjustment of a single
multi-millennial mode. However, we stress that the response to $\Omega'$ is *not* simply the response of a single multi-decadal mode,
because the grounding zone is coupled to the slower, but ultimately more sensitive interior reservoir.

While these mathematical interpretations may seem abstract, it is helpful to remember that the two-stage model was derived
from mass conservation, and that the linear response times (eigenmodes) reflect the large-scale glacier geometry. Because a
glacier is a Stokes (i.e., non-inertial) fluid driven by potential-energy gradients, the large-scale glacier geometry must reflect
the aggregate dynamics by which the system seeks flux balance. This relationship between geometry and dynamics is another
way to interpret why $\Omega'$ and $S'$ forcings project differently onto the fast and slow modes: these flux imbalances have different
geometries, and thus the geometry and dynamics of the transient responses must also be distinct.

To help illustrate this, Fig. 3 tracks the evolution of fluxes following step changes in $S$ and $\Omega$. Five fluxes are shown for the
flowline model: over the interior surface ( $S \cdot L$), across the grounding line ($Q_g$), and at conceptual flux gates located ~~5,
25, and 50$S \times L$~~ $S \cdot L$, $Q$, and
$Q_g$ are shown (Fig. 3b). In all cases,  flux anomalies are normalized by their  final (equilibrium)
change. A change in surface mass balance has an immediate, spatially distributed tendency on interior thickness, which slowly
alters driving stresses and thus fluxes throughout the interior. Anomalous ice flux arrives at the grounding zone, driving advance
or retreat, which then modifies $Q_g$ according to  Equation (1). Most of the disequilibrium during the transient response is
between surface mass balance and interior fluxes, while $Q_g$ can quickly keep pace with flux anomalies from the interior. In the
two-stage model, $Q_g$ keeps up on the fast timescale. In contrast, a perturbation to $Q_g$ (here, via $\Omega$) is highly localized and first
creates a large disequilibrium between $Q_g$ and interior flux. The increase in $Q_g$ causes the grounding zone to drain, drawing
in ice from the interior, which transfers the disequilibrium from $Q$ and $Q_g$, to $Q$ and  $S \cdot L$. Both models capture this
transfer, and the flowline model flux gates show that it gradually propagates up from the grounding zone. The transient peak in
fluxes is similar to a kinematic wave propagating from the terminus, which reaches well into the interior within a few multiples

[revised manuscript text omitted]
 (no persistence; black);  red noise with $\tau_{AR1} = 4$ yr  (red)  and $\tau_{AR1} = 20$ yr  (maroon); and power-law (noise with $\nu = 0.5$  (blue).  Time series shown are dimensionless and offset for clarity, with vertical ticks of $1\sigma$. **(b)** The glacier-length variations generated by the two-stage model, when the synthetic anomalies are applied as $\Omega'$ (dark lines) and $S'$ (pastel lines).  Time series are again offset, with vertical ticks of 1 km. **(c)** Power spectra of the synthetic climate forcings. For reference, vertical lines correspond to  $\frac{1}{\tau_F}$ and  $\frac{1}{\tau_S}$. **(d)** Spectra of the glacier's response to white noise, which illustrates the glacier's sensitivity as a function of frequency, and thus its temporal filtering properties. This depends on whether forcing is applied  as $\Omega'$ (black) or  $S'$ (gray). **(e)** Spectra of glacier response to  $\Omega'$ (dark lines) or $S'$ (pastel lines). Responses depend on the spectra of the forcing and the glacier's filtering properties. Forcing with persistence has more power at low frequencies where glacier sensitivity is highest, increasing the overall glacier variance.

To illustrate the current committed change of outlet glaciers, we use the two-stage model and follow the framework of Christian et al. (2018), using idealized glacier geometries and forcings. Disequilibrium, and thus committed retreat, is defined in reference to an "instantaneous equilibrium" response, which is the state at which the glacier would be in equilibrium with climate at any given time. This is governed largely by the bed geometry and climate forcing. An idealized geometry (e.g., uniform bed slope and width) makes for a simple equilibrium response, but the physical principle of course also holds for more complex systems. For an idealized outlet glacier, the instantaneous equilibrium is the grounding-line position ($L$) that yields $Q_g = S \cdot L$.

We consider ramp forcing scenarios where $S$ decreases, or $\Omega$ increases, by 30% from 1880 to 2020 CE. The forcing is fixed at 2020, and the glacier is allowed to relax towards a new equilibrium. Figure 5 shows the responses of three glaciers with varied response times: (1) $\tau_F, \tau_S \sim (76, 2000)$ yr; (2) $\tau_F, \tau_S \sim (56, 1200)$ yr; and (3) $\tau_F, \tau_S \sim (140, 4600)$ yr. The parameters for each glacier are provided in Table 2. Dashed lines show the instantaneous equilibrium responses, which indicate the total committed response at any given time.

The most basic result is that, in all cases, the transient response as of 2020 is a small fraction ($\sim 1$–$23\%$) of the instantaneous equilibrium response. The disequilibrium for $S$ forcing is particularly severe, as pointed out previously by RRH. Although changes in surface mass balance are already a major component of overall mass loss in some settings, basic glacier dynamics require that $S$ anomalies also have an impact on the grounding-line position, which is, as yet, essentially unrealized. Perturbations in $\Omega$ elicit a faster response over the industrial era, but Fig. 5a–c shows that both types of forcing have a long-term commitment associated with the slow drawdown of interior ice.

The "slow response" (interior thinning $\to$ reduced interior fluxes $\to$ grounding-line retreat) comprises the majority of the response to $S$ forcing, but it is important to remember that it is also the second stage of the response to ocean forcing, and it dictates the total committed change. Following recent observed increases in ice discharge, Price et al. (2011) and Goldberg et al. (2015) examined the committed responses of several outlet glaciers in Greenland and Antarctica, respectively. These studies found substantial "dynamic" commitments associated with thinning and elevated fluxes, although they focused on near-term sea-level rise and did not project past 2100. Our flux-balance perspective is a complementary view of committed change, and shows that dynamic thinning necessarily drives additional retreat on long timescales as a consequence of reduced interior fluxes. The slow equilibration of the interior would thus be a critical part of determining the total commitment due to ocean forcing, especially if the long-term retreat moves the terminus to more- or less-stable slopes.

The slow mode also means that there can be large uncertainties in committed change if the magnitude of forcing is uncertain. Figure 5d shows the response to a trend in $\Omega$, ranging from a $20$–$40\%$ increase from the initial value. As of 2020, the differences in observed retreat are small, but diverge as the system approaches equilibrium over subsequent millennia. In other words, the slow response limits how "wrong" short-term simulations might be due to errors in the assumed forcing, but makes no such constraints on the committed change. The same principle also applies to uncertainty in the outlet glacier's length sensitivity: note that glaciers 1 and 2 have nearly identical responses on centennial timescales, but different equilibrium sensitivities. Similar arguments have been made for uncertainty in radiative forcing and equilibrium climate sensitivity (Armour and Roe, 2011).

**Table 2.** Parameters varied between three idealized glaciers (top) and the resulting steady-state values (bottom). The steady state is calculated by the nonlinear two-stage model, and the linearized response times are given by Equations (13) and (14).

| Parameter | Description | Units | Glacier 1 | Glacier 2 | Glacier 3 |
|---|---|---|---|---|---|
| $\bar{S}$ | Surface mass balance | m yr$^{-1}$ | 0.5 | 0.6 | 0.3 |
| $\Theta$ | Buttressing factor | | 0.7 | 0.75 | 0.6 |
| $b_0$ | Bed elev. at divide | m | $-100$ | $+150$ | $+100$ |
| $b_x$ | Bed slope | | $-2 \times 10^{-3}$ | $-3 \times 10^{-3}$ | $-1 \times 10^{-3}$ |
| **Steady-state value** | | | | | |
| $\bar{H}$ | Interior thickness | m | 1413 | 1569 | 2814 |
| $\bar{L}$ | Length | km | 185 | 212 | 700 |
| $h_g$ | Grounding line thickness | m | 526 | 545 | 673 |
| $\tau_F$ | Fast response time | yr | 77 | 56 | 144 |
| $\tau_S$ | Slow response time | yr | 2030 | 1160 | 4590 |

**3.2 The emergence and detectability of forced responses**

We now turn to the topic of attributing outlet glacier retreats to natural or anthropogenic forcing. The attribution of an observed change to a particular cause (i.e., an external forcing) can be a challenge because of factors specific to individual glaciers, such as complexities in bed geometry, regional climate, or the local collection of ice-ocean interactions. It can also be a challenge because of factors intrinsic to the transient ice dynamics, which affect the amount of the forced response that can be expressed over a given time. 
[revised manuscript text omitted]
 natural variability are considered in each panel:  The top row corresponds to interannual variability in $S$ ; middle row to interannual variability in $\Omega$ ; and bottom row to multidecadal variability ( $\tau_{AR1} = 20$ yr) in $\Omega$ . **(a)**  The idealized climate trends (plus variability) beginning in 1880. In all cases, the linear trend reaches a SNR of  1 by 2020. **(b)** Grounding-line responses to each idealized trend. Shaded regions are the  $1\sigma_L$ and  $2\sigma_L$ bounds for each type of noise. The orange lines indicate when the trend has been applied. Thinner lines show the grounding-line response without variability. **(c)** Probability density functions for grounding-line trends driven  by each type of natural variability, but no external forcing, over time periods from  50–500 years. Note the different length scales in each case. Trends on the order of km century$^{-1}$ are extremely unlikely to occur due to variability in $S$ alone (top), but commonplace if the glacier is sensitive to multidecadal variability in $\Omega$ (bottom).

variability has significant persistence. The sensitivity of grounding line flux to ocean variability and large calving events, and the associated timescales, will thus be critical for attribution studies. Finally, it should be borne in mind that these detection challenges arise from slow dynamics, and not from low sensitivity. As is clear from Fig. 5, only a small fraction of the committed response is available for attribution studies today.

**3.3 Inherited conditions**

We have thus far considered outlet-glacier fluctuations due to stationary interannual-to-multidecadal climate variability. However, long response times also imply some memory of climate variations throughout the Holocene. Ice-sheet models are often

"spun up" using paleoclimate proxy data precisely for this reason (e.g., Bindschadler et al., 2013). However, these strategies do not always reproduce the same modern ice extent (e.g., Goelzer et al., 2018), and the simulated ice-sheet history can depend strongly on the choice of proxy and its implementation in the model (e.g., Nielsen et al., 2018; Buizert et al., 20 . Here, we compare outlet glacier responses to ocean versus interior forcing over the Holocene as an additional factor for the

470   modern state. We focus on climate anomalies that are well-documented in the Northern Hemisphere, but similar considerations would apply to Antarctic outlet glaciers.

First, we consider a climate scenario with idealized representations of three events: a deglacial warming at  11 kya, a Little Ice Age (LIA) cool period from 1450 to 1850 CE, and an anthropogenic warming trend from 1880 to  2100 CE. The deglacial and LIA transitions are smoothed by error functions of  500 yr and 50 yr widths, respectively.

475   The magnitudes of deglacial, LIA, and anthropogenic events have a ratio of 10:1: 4, and the intervening Holocene climate is assumed constant. We scale forcings linearly to the climate anomalies, where warming corresponds to negative $S'$  or positive $\Omega'$.  Obviously, different combinations of these forcings could be expected in reality. Rather

480   than choosing a particular combination, we examine each in isolation and normalize the glacier responses. These experiments thus serve as limiting cases to illustrate the relative influences of ocean and interior forcing over different timescales. This scenario is designed to explore two practical points: (1) the glacier's memory of large, long-ago events compared to smaller, more recent events; and (2) the glacier's relative memory of past ocean vs. interior forcing.

We use the two-stage model, linearized with respect to the Holocene climate with parameters for glacier 1 (See Table 2;

485   $\tau_F, \tau_S \sim 76, 2000$ yrs). The advantage of using the linear model here is that it has uniform sensitivity to fractional perturbations in $S$ and $\Omega$. This allows us to more clearly distinguish the signatures of fast and slow dynamics; the tradeoff is that it ignores nonlinearities that are surely a factor for very large climate changes. Accurate simulations over such transitions would depend not only on nonlinearities in ice dynamics, but also on spatial information (e.g., bed topography) that is simplified in reduced models. Nevertheless, the linear model is a straightforward tool for demonstrating consequences of having both fast and slow

490   dynamics. Focus should be directed toward the responses to the idealized LIA, for which the linearization is more valid. Including the deglaciation signal primarily serves to account for residual, albeit faint, disequilibrium implied by millennial response times.

Figure 7a shows the idealized climate (top) and two-stage model responses. Anomalies are shown relative to the mid-Holocene and normalized to the large deglacial transition. Figure 7b shows 1000 to

495   2000 CE in more detail, including a scenario with no anthropogenic warming (dashed).

 For reference, the length at which the glacier would be in equilibrium with the LIA climate is also plotted (gray). For both forcings, the grounding-line retreat due to the deglacial signal is  nearly complete by the onset of the LIA. However, the  LIA advance is $\sim 2\times$ greater for forcing in $\Omega$, because it can engage the fast mode to a greater degree. Yet, even forcing in $\Omega$ yields a muted response: the

500   transient

response only reaches $\sim 35\%$ equilibration before the period ends. The slow mode, which takes up the majority of the response for both forcing types, barely feels our  400 yr LIA before it is reversed. This is worth bearing in mind whenever the duration of glacier excursions and climate anomalies are less than $\tau_S$, because the system never achieves equilibrium. This would be an issue particularly if such events are used to tune glacier sensitivity in models.

505     The idealized LIA is a much more discrete "event" than is supported by paleoclimate records, and ignores other variations in the Holocene. Thus, we also consider the response to a more realistic forcing  time series. Again, this is not  a reconstruction of actual terminus changes, but it is useful to see how glacier memory integrates the continuum of variations found in paleoclimate records. We use a  time series of temperatures for Disko Bay, Greenland, from the regional reconstruction of  Buizert et al. (2018) (Fig. 7c–d). The forcings are scaled as 510    above, and normalized to 1880 CE.

The glacier responses in Fig. 7d show the same essential behavior as the more idealized case in Fig. 7b. Even with a more subtle LIA, the terminus response is much more pronounced if it is driven by $\Omega$ anomalies that engage the fast response. It is worth noting, though, that most of the glacier disequilibrium in the 1800s is due to cooling over the previous millennia, to which the slow mode responds with a pronounced lag. Thus, the preindustrial state of an outlet glacier may depend significantly 515    both on LIA and longer-term ocean cooling, via its two distinct response timescales.

**4    Summary and discussion**

Marine-terminating outlet glaciers are sensitive to two fundamental types of forcing: changes in surface mass balance, which  are distributed over a large interior catchment; and changes in their flux at the grounding line, which are typically driven by the ocean. We have used two models of different complexity to explore and contrast the dynamic responses to these two categories 520    of forcing. Our key findings are as follows:

1. Ocean forcing (via the grounding-line flux coefficient, $\Omega$) and interior surface-mass-balance forcing elicit fundamentally different transient grounding-line responses (Figs. 1 and 2). The response to ocean forcing is characterized by a fast initial grounding-line migration, followed by a second, much slower stage of adjustment. In contrast, the response to interior forcing is dominated by slow grounding-line migration, with very strong damping at short timescales.

525    2. The two-stage model  (Robel et al., 2018) captures the evolution of flux anomalies that gives rise to these two contrasting dynamic responses (Fig. 3). This lends further confidence to the two-mode interpretation of outlet glacier dynamics: a fast mode associated with adjustment of the grounding zone, and a slow mode associated with the interior reservoir. We have shown here that  ocean forcing projects onto both modes, while interior forcing projects almost entirely onto the slow mode.

530    3. Persistence in stochastic climate forcing amplifies natural grounding-line fluctuations (Fig. 4). Increased persistence means more of the variance in the forcing occurs at low frequencies, where the glacier is ultimately more sensitive. In particular, multidecadal ocean variability can drive large fluctuations on multidecadal to millennial timescales. Under-

[Figure]

**Figure 7.** Response to long-term climate variations. **(a)** An idealized climate scenario representing the last deglaciation, Little Ice Age (LIA), and anthropogenic forcing (black line). Normalized length responses of the linear model are shown for forcing applied via $S$ (brown) and $\Omega$ (blue). The length at which the glacier would remain in equilibrium with climate is shown for reference (gray). **(b)** As for **(a)**, but zoomed into the last 1000 yr. $\Omega$ forcing  produces a much larger response  on multi-century timescales, but the  glacier does not reach equilibrium with the LIA in either case. **(c)** A more realistic forcing time series, 
[revised manuscript text omitted]